# Pruning as a Cooperative Game: Surrogate-Assisted Layer Contribution Estimation for Large Language Models

**Xuan Ding**
Shenzhen Future Network of Intelligence Institute
Guangdong Provincial Key Laboratory of Future Networks of Intelligence
The Chinese University of Hong Kong (Shenzhen)
202322011119@mail.bnu.edu.cn

**Pengyu Tong**
Beijing Normal University
202522011202@mail.bnu.edu.cn

**Ranjie Duan**
Alibaba Group
ranjieduan@gmail.com

**Yunjian Zhang**
University of Chinese Academy of Sciences
sdtczyj@gmail.com

**Rui Sun** [*]
Shenzhen Future Network of Intelligence Institute
Guangdong Provincial Key Laboratory of Future Networks of Intelligence
The Chinese University of Hong Kong (Shenzhen)
ruisun@link.cuhk.edu.cn

**Yao Zhu** [*]
Zhejiang University
ee_zhuy@zju.edu.cn

## Abstract

While large language models (LLMs) demonstrate impressive performance across various tasks, their deployment in real-world scenarios is still constrained by high computational demands. Layer-wise pruning, a commonly employed strategy to mitigate inference costs, can partially address this challenge. However, existing approaches generally depend on static heuristic rules and fail to account for the interdependencies among layers, thereby limiting the effectiveness of the pruning process. To this end, this paper proposes a game-theoretic framework that formulates layer pruning as a cooperative game in which each layer acts as a player and model performance serves as the utility. As computing exact Shapley values is computationally infeasible for large language models (LLMs), we propose using a lightweight surrogate network to estimate layer-wise marginal contributions. This network can predict LLM performance for arbitrary layer combinations at a low computational cost. Additionally, we employ stratified Monte Carlo mask sampling to further reduce the cost of Sharpley value estimation. This approach captures inter-layer dependencies and dynamically identifies critical layers for pruning. Extensive experiments demonstrate the consistent superiority of our method in terms of perplexity and zero-shot accuracy, achieving more efficient and effective layer-wise pruning for large language models. The code is available at: `https://github.com/920927/Pruning_As_A_Cooperative_Game`.

---

[*]corresponding author

# 1 INTRODUCTION

Large language models (LLMs) achieve state-of-the-art performance across a wide range of tasks (Chen et al., 2023; Duan et al., 2024; Zhu et al., 2025), but their massive computational and memory requirements pose significant challenges for practical deployment (Wang et al., 2024a; Sun et al., 2024a). This has prompted extensive research into model compression techniques. Among these, layer pruning, which removes entire transformer layers, stands out as an effective method for reducing inference cost. Compared to width pruning, depth pruning offers superior throughput and inference speed, while also being easier to implement than other compression methods, making it an attractive approach for large-scale model compression (Kim et al., 2024).

Existing deep pruning methods typically assign an importance score to each layer to determine the pruning order. These scores are often based on heuristics such as weight magnitudes, activation norms (Filters'Importance, 2016), or sensitivity analysis (Men et al., 2024; Kim et al., 2024), assuming that layer importance is fixed and independent. However, our experiments reveal that layer importance is context-dependent. In single-layer pruning (Fig. 1a), the rankings of early and late layers remain relatively stable, whereas the rankings of middle layers fluctuate significantly. When extended to multi-layer pruning (Fig. 1b), this volatility is further amplified, with some layers showing strong fluctuations throughout the pruning process. These observations highlight the dynamic interdependencies between layers: pruning one layer can alter the relative importance of others, and evaluating layers in isolation often results in suboptimal pruning decisions. Previous studies have partially considered interactions between layers, such as merging redundant layers (Ding et al., 2025b) or progressively pruning less important ones during training (Song et al., 2024). However, these approaches often fail to find the globally optimal layer set. For example, pruning layers sequentially based on individual importance may not yield the best two-layer combination, because the optimal strategy could involve simultaneously removing a pair of layers that are not the least important individually (see Tab. 1).

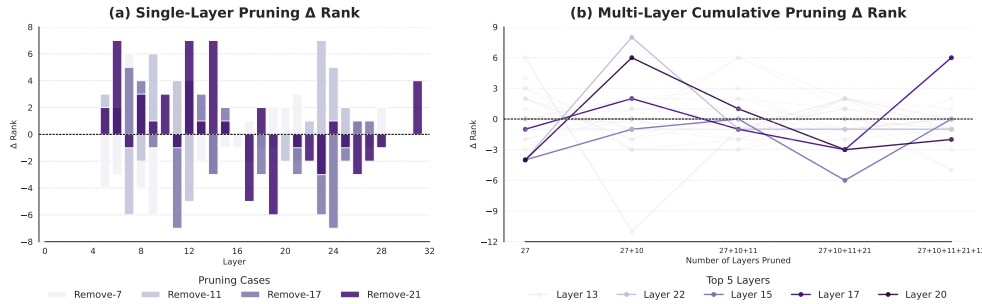

Figure 1: **Layer importance is context-dependent under pruning.** Both plots are based on the change in PPL before and after pruning on the BookCorpus dataset to rank layer importance. (**Left**) Bar plot showing the $\Delta$Rank changes for random single-layer pruning, highlighting that some layers' importance shifts more dramatically when others are pruned. (**Right**) Line plot showing the $\Delta$Rank changes during multi-layer pruning, where the lowest-ranked layer is removed at each step. The five most volatile layers are highlighted in darker colors, reflecting their fluctuating importance.

| Scheme | Deleted Layers | Post-deletion PPL | Explanation |
|---|---|---|---|
| | Layer 27 | 14.9845 | Delete the least important layer |
| Single Deletion | Layer 10 | 14.9915 | Delete the second least important layer |
| | Layer 11 | 15.0154 | Delete the third least important layer |
| Scheme 1 | Layer 27 + Layer 10 | 15.4535 | Delete the two least important layers |
| Scheme 2 | Layer 27 + Layer 10 | 15.4535 | Delete the least important layer, re-test, then delete the next |
| Optimal Combo | Layer 10 + Layer 11 | 15.4279 | Delete a pair of layers accounting for inter-layer interactions |

Table 1: **Summary of layer deletion schemes and their corresponding PPL values on Book-Corpus.** Both pruning based on static importance and re-calculated importance after each deletion may not always lead to optimal performance.

To address the limitations of static heuristics and explicitly capture the dynamic interdependencies among layers, we formulate LLM pruning as a cooperative game, where each Transformer layer is treated as a player with the model's performance defining the utility. In this setting, a layer's contribution is inherently context-dependent, shaped by its interactions with other layers. While Shapley values offer a principled way to quantify such contributions, their exact computation is intractable for large-scale models due to the exponential number of possible layer combinations. To make this feasible, we propose a two-stage approximation strategy aligned with cooperative game theory. In the first stage, we generate diverse pruning masks through stratified Monte Carlo sampling with controlled Hamming weights, and evaluate them on calibration data to measure perplexity (PPL). The performance gap between each pruned model and the original model provides supervision signals for learning. In the second stage, we train a lightweight surrogate network to predict these performance drops for unseen masks, enabling efficient estimation of marginal contributions without repeated full-model evaluations. Once trained, the surrogate allows us to estimate Shapley values from a large pool of candidate masks. This design preserves inter-layer dependencies, adaptively identifies critical layers, and scales effectively to pruning large language models.

Extensive experiments on language modeling and downstream tasks demonstrate the effectiveness of our method. Specifically, we report perplexity on WikiText, PTB, and C4, and assess inference performance across eight zero-shot benchmarks and an adversarial reasoning robustness metric. Compared to depth-wise and width-wise pruning baselines, our method achieves lower perplexity, higher accuracy, and favorable trade-offs in speed and throughput. Furthermore, we show that our framework extends beyond Transformer-based LLMs, demonstrating strong generality on non-Transformer architectures, and can be seamlessly combined with quantization to deliver additional efficiency gains.

In summary, our contributions are as follows:

- We rethink LLM pruning from a game-theoretic perspective, treating layers as interdependent players and revealing inter-layer dependencies that static heuristics fail to capture.
- We propose a scalable approximation framework that leverages stratified Monte Carlo mask sampling and a lightweight surrogate network, enabling efficient Shapley-based estimation of layer contributions in large LLMs.
- We validate our method on language modeling tasks and zero-shot benchmarks, showing consistent improvements over strong pruning baselines across diverse architectures.

## 2 RELATED WORK

### 2.1 PRUNING METHODS FOR LARGE LANGUAGE MODELS

The rapid growth of large language models (LLMs) has led to the development of various compression techniques, including quantization (Frantar et al., 2023; Dettmers et al., 2022), knowledge distillation (Fu et al., 2023; Hsieh et al., 2023), tensor decomposition (Wang et al., 2024b; Ding et al., 2025a), and pruning. Pruning reduces inference costs by removing model components without complex retraining. It has evolved from unstructured sparsity (removing individual weights) to structured sparsity (pruning entire neurons, attention heads, or layers). For example, SparseGPT uses the OBS error formula to assess weight importance and decide whether to prune, while addressing the challenge of non-structured sparsity on real hardware through semi-structured pruning (Frantar & Alistarh, 2023). Methods like Wanda combine weight magnitudes with input feature norms for layer selection (Sun et al., 2023). LLM-Pruner and FLAP focus on pruning coupled structures, such as attention heads, to reduce network width while maintaining the number of layers (Ma et al., 2023; An et al., 2024). These methods show pruning's potential for deploying LLMs on resource-constrained devices with minimal performance loss.

### 2.2 MEASURING LAYER CONTRIBUTIONS

Direct layer-wise pruning methods are straightforward and effective, offering better inference speed and throughput. Most of these methods in LLM pruning use heuristics like weight magnitude, gradient sensitivity, or activation statistics to assess layer importance. ShortGPT introduces Block Influence (BI) to quantify the importance of each layer and prunes redundant layers (Men et al., 2024),

while LAYERIF tracks the sensitivity of different layers to training data using influence function (Askari et al., 2025). Shortened-LLaMA, on the other hand, calculates weight importance scores at the output neuron level using Taylor+ and PPL metrics, assessing block-level importance (Kim et al., 2024). SLEB prunes redundant transformer blocks iteratively using Metric3, integrating smoothly into the forward pass (Song et al., 2024). In contrast, CALM, Mixture-of-Depths, and SkipDecode dynamically allocate computation resources based on context, adjusting compute expenditure to optimize efficiency (Del Corro et al., 2023; Raposo et al., 2024; Schuster et al., 2022).

Cooperative game theory studies how multiple rational decision-makers form alliances to achieve common goals. This makes it well-suited for assessing the importance of different layers in LLMs. The GTAP method, for instance, treats neurons as cooperative agents and uses power indices to evaluate importance (Diaz-Ortiz Jr et al., 2023). While computational complexity limits its application to ultra-large models, the principles offer a valuable perspective for pruning. Zhang et al. (2024) applied the Shapley value to model interpretation, noting its computational infeasibility for LLMs. They proposed using early truncation or similar SV-NUP methods, considering only the influence of adjacent layers for non-uniform pruning (Sun et al., 2025).

## 3 METHOD

### 3.1 LAYER PRUNING AS A COOPERATIVE GAME

Layer pruning in large language models (LLMs) is inherently challenging because the contribution of each Transformer layer is not independent: conventional importance rankings often ignore inter-layer dependencies, leading to suboptimal pruning. To address this, we formulate layer pruning as a *cooperative game*, where each layer acts as a *player* and the model's performance defines the *utility function*.

Formally, let $\mathcal{L} = \{1, 2, \ldots, L\}$ denote the set of layers in an LLM with $L$ layers. For any subset of layers $S \subseteq \mathcal{L}$, let $M(S)$ be the model obtained by retaining only layers in $S$, and let $u(S)$ denote its performance measured by perplexity (PPL) on calibration data, where a lower PPL corresponds to higher utility. The *marginal contribution* of layer $i$ to a subset $S$ is

$$\Delta_i(S) = u(S \cup \{i\}) - u(S). \tag{1}$$

While the Shapley value provides a theoretically grounded measure of layer importance, computing it exactly is infeasible due to the exponential number of subsets. This motivates our *two-stage approximation framework*, which efficiently estimates layer contributions while preserving inter-layer dependencies.

### 3.2 ALGORITHM OVERVIEW

Figure 2 illustrates the overall pipeline of our approach. The goal is to efficiently estimate the contribution of each layer to the model's performance, enabling informed pruning decisions while preserving inter-layer dependencies. Our framework proceeds in two stages:

1. **Mask Generation and Performance Evaluation:** In the first stage, we generate diverse pruning masks using stratified Monte Carlo sampling with controlled Hamming weights. Each mask defines a subset of layers to retain, and we evaluate the corresponding pruned models on calibration data to obtain performance scores (perplexity differences). These scores serve as supervision signals for learning.

2. **Surrogate Training and Shapley Value Estimation:** In the second stage, we train a lightweight surrogate network $f_\theta$ to predict the performance of unseen masks. Once trained, the surrogate enables efficient estimation of each layer's marginal contribution, which is then aggregated to compute approximate Shapley values. Finally, layers with the lowest contributions are pruned.

For clarity, the complete procedure is also provided in pseudo-code in Algorithm 1 in Appendix. B.3.

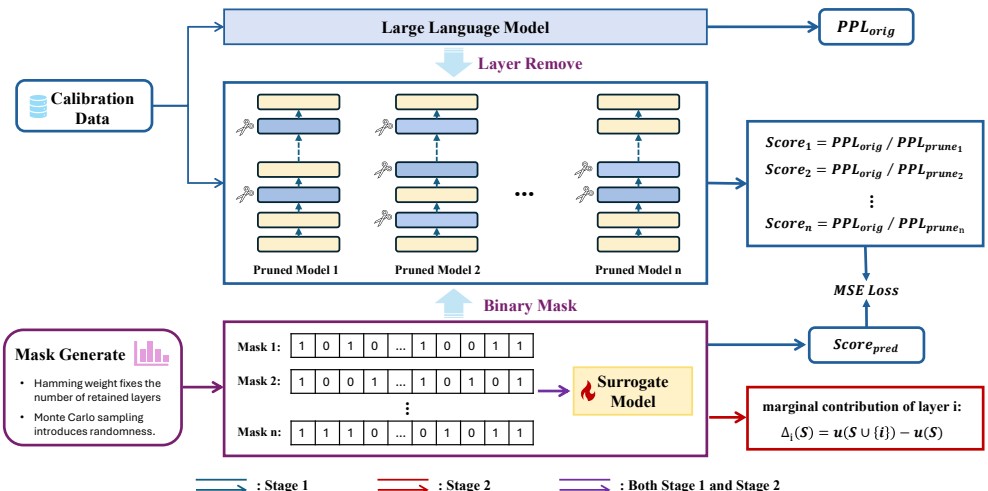

Figure 2: **Framework of our method.** Both stages use stratified Monte Carlo masks with controlled Hamming weight. Stage one uses calibration data to compute PPL-based scores for training a lightweight surrogate network, and stage two uses the surrogate to efficiently compute Shapley-based layer importance for scalable LLM pruning.

### 3.3  STAGE ONE: MASK GENERATION AND PERFORMANCE EVALUATION

Directly computing Shapley values requires evaluating all $2^L$ layer subsets, which is computationally infeasible for LLMs. In the first stage, we approximate layer contributions via stratified Monte Carlo sampling of binary pruning masks. Each mask $\mathbf{m} \in \{0, 1\}^L$ denotes a subset of retained layers, where $\mathbf{m}_i = 1$ indicates that layer $i$ is preserved. The corresponding pruned model is $M(\mathbf{m})$.

To quantify the contribution of each mask, we compute a performance score as the ratio of the original perplexity to the pruned model's perplexity:

$$s(\mathbf{m}) = \frac{\text{PPL}_{\text{orig}}}{\text{PPL}(M(\mathbf{m}))}, \tag{2}$$

where $s(\mathbf{m})$ closer to 1 indicates better preservation of the original model's performance.

To ensure balanced exploration across pruning ratios, we design a stratified sampling strategy based on Hamming weight (i.e., number of retained layers $k(\mathbf{m}) = \sum_{i=1}^{L} m_i$). Let $\mathcal{K} = \{k_1, \ldots, k_{|\mathcal{K}|}\}$ be the set of target weights. For each weight $k_j \in \mathcal{K}$ we draw $N_{k_j}$ masks that retain exactly $k_j$ layers. Given a total budget of $N$ masks we enforce $\sum_{j=1}^{|\mathcal{K}|} N_{k_j} = N$ and in practice set $N_{k_j} \approx \lfloor N/|\mathcal{K}| \rfloor$ (distributing any remainder to the first few strata). The $t$-th mask sampled within the stratum of Hamming weight $k_j$ is denoted by $\mathbf{m}^{(k_j, t)}$ and drawn as

$$\mathbf{m}^{(k_j, t)} \sim \text{Uniform}\Big\{\mathbf{m} \in \{0, 1\}^L : k(\mathbf{m}) = k_j\Big\}, \quad t = 1, \ldots, N_{k_j}, \ j = 1, \ldots, |\mathcal{K}|. \tag{3}$$

This stratified sampling ensures balanced coverage across pruning ratios, while Monte Carlo randomness captures diverse interactions among layers. The resulting masks and their corresponding scores form the training data for the surrogate network in Stage Two.

### 3.4  STAGE TWO: SURROGATE TRAINING AND SHAPLEY VALUE ESTIMATION

Direct evaluation of $s(\mathbf{m})$ for every pruned model is computationally prohibitive. To address this, we introduce a lightweight surrogate network $f_\theta(\mathbf{m})$, implemented as a two-layer feed-forward network, that predicts the performance score of any binary mask $\mathbf{m}$. This surrogate decouples expensive full-model inference from large-scale Shapley value estimation.

**Training the surrogate** The surrogate is trained on the limited set of masks and their true scores obtained in Stage One, denoted as $\{(\mathbf{m}^{(k_j,t)}, s(\mathbf{m}^{(k_j,t)}))\}$. We optimize the mean squared error:

$$\mathcal{L}(\theta) = \frac{1}{N} \sum_{n=1}^{N} \left( f_\theta(\mathbf{m}_n) - s(\mathbf{m}_n) \right)^2, \tag{4}$$

where $N$ is the total number of training masks. Once trained, $f_\theta$ generalizes to unseen masks, enabling efficient prediction of model performance without costly full-model evaluations.

**Approximating Shapley values** Using the surrogate, we approximate each layer's Shapley value via stratified Monte Carlo sampling in Eq. (3). For layer $i$, we repeatedly sample $Q$ binary masks $\mathbf{m}^{(k_j,q)}$ and compute the marginal contribution of layer $i$:

$$\hat{\phi}_i = \frac{1}{Q} \sum_{q=1}^{Q} \left( f_\theta(\mathbf{m}^{(k_j,q)} \cup \{i\}) - f_\theta(\mathbf{m}^{(k_j,q)}) \right). \tag{5}$$

This efficiently estimates layer-wise contributions while preserving inter-layer dependencies, as the surrogate captures performance shifts under diverse layer coalitions.

**Layer pruning** Finally, layers are ranked by their estimated Shapley values $\{\hat{\phi}_i\}_{i=1}^{L}$. We remove the least-contributing layers until the target compression ratio. The resulting pruned LLM retains critical layers while eliminating redundant ones, maintaining overall model performance.

## 4 EXPERIMENTS

### 4.1 EXPERIMENTAL SETUP

**Foundation LLMs.** We evaluate on open-source LLMs, including Transformer models (LLaMA2-{7B, 13B} (Touvron et al., 2023), Meta-LLaMA3-8B, Vicuna-7B-v1.3 (Chiang et al., 2023)) and non-Transformer models (RWKV-7B (Peng et al., 2023), Mamba-2.8B (Gu & Dao, 2023)).

**Benchmarks.** Model performance is assessed on language modeling and zero-shot reasoning tasks. For language modeling, we measure perplexity on WikiText2, PTB, and C4 to quantify generative quality after pruning. Zero-shot reasoning is evaluated on nine datasets: PIQA (Bisk et al., 2020), HellaSwag (Zellers et al., 2019), ARC-easy and ARC-challenge (Clark et al., 2018), OpenbookQA (Mihaylov et al., 2018), RACE (Lai et al., 2017), WSC273 (Levesque et al., 2012), LAMBADA (Paperno et al., 2016), and MMLU (Hendrycks et al., 2021), using the `lm-evaluation-harness` (Gao et al., 2024). We further test adversarial reasoning robustness on ANLI (Nie et al., 2020) across its three rounds (R1–R3).

**Baselines.** We compare against width pruning methods (LLM-Pruner (Ma et al., 2023), Wanda-sp, FLAP (An et al., 2024)) and depth pruning methods (SliceGPT (Ashkboos et al., 2024), SLEB (Song et al., 2024), Shortened-LLM (Kim et al., 2024), ShortGPT (Men et al., 2024)).

**Implementation Details.** Experiments are implemented in PyTorch (Paszke et al., 2019) using HuggingFace Transformers (Wolf et al., 2020). Following Ma et al. (2023), we randomly sample 10 BookCorpus (Zhu, 2015) examples for pruning calibration, using the same set across all baselines for fair comparison. We provide additional experiments on LoRA finetuning in Appendix. E, ablation studies in Appendix. F, and an analysis of the computational cost of our method in Appendix. G.

### 4.2 LANGUAGE MODELING

Tab. 2 compares compares the language modeling performance of our method with depth-wise pruning baselines. Our approach consistently yields the lowest or near-lowest perplexity across models and pruning ratios, with the advantage becoming more pronounced under aggressive pruning. Notably, while baseline methods on Meta-LLaMA-3-8B degrade sharply at high pruning ratios, our method maintains stable and significantly lower perplexity, confirming its effectiveness in preserving generative performance under substantial compression.

| Method | Remove 3 layers | | | Remove 6 layers | | | Remove 9 layers | | | Remove 12 layers | | |
|---|---|---|---|---|---|---|---|---|---|---|---|---|
| | PPL_WikiText2 | PPL_PTB | PPL_C4 | PPL_WikiText2 | PPL_PTB | PPL_C4 | PPL_WikiText2 | PPL_PTB | PPL_C4 | PPL_WikiText2 | PPL_PTB | PPL_C4 |
| **LLaMA-2-7B-hf** | | | | | | | | | | | | |
| SliceGPT | 108.0990 | 131.3884 | 103.9473 | 212.8867 | 219.2298 | 191.5134 | 291.8482 | 293.3186 | 257.7473 | 393.8880 | 365.7072 | 343.4214 |
| SLEB | **14.2428** | **52.9183** | 12.9682 | 19.4676 | 63.8317 | 16.3933 | 27.4537 | 79.4398 | 21.3809 | 58.1194 | 135.1317 | 43.8708 |
| Shortened-LLaMA | 16.6515 | 54.5982 | 13.8046 | 36.3702 | 105.2407 | 29.2243 | 81.9615 | 196.6155 | 61.8678 | 304.5240 | 486.6280 | 252.4593 |
| ShortGPT | 16.6515 | 54.5982 | 13.5906 | 36.3702 | 105.2407 | 29.2243 | 81.9615 | 196.6155 | 61.8678 | 157.9850 | 295.1548 | 98.8645 |
| Ours | 14.6949 | 53.7517 | 12.9682 | 18.8686 | 61.8678 | **16.1392** | 24.6093 | 76.9957 | 20.7231 | **38.1157** | 105.2407 | 28.7712 |
| **Vicuna-7B-v1.3** | | | | | | | | | | | | |
| SliceGPT | 151.4702 | 195.3507 | 133.0439 | 292.3765 | 339.8809 | 250.8779 | 401.3294 | 435.8758 | 343.5819 | 555.5617 | 566.6461 | 474.9028 |
| SLEB | **19.7741** | 74.6268 | **16.1392** | 26.6090 | 87.2476 | 21.3809 | 38.1157 | 115.5843 | 30.6268 | 65.8579 | 196.6155 | 47.4357 |
| Shortened-LLaMA | 20.4018 | 70.1054 | 17.4506 | 35.8063 | 98.8645 | 27.8860 | 67.9485 | 143.8470 | 48.1827 | 244.6919 | 356.0247 | 157.9850 |
| ShortGPT | 23.1183 | 79.4398 | 18.0046 | 67.9485 | 143.8470 | 48.1827 | 67.9485 | 143.8470 | 48.1827 | 252.4593 | 518.0128 | 153.1243 |
| Ours | 20.7231 | **70.1054** | 17.7254 | 24.6093 | 81.9615 | 20.7231 | 37.5247 | 112.0281 | 29.2243 | 67.9485 | 209.2961 | **43.1907** |
| **Meta-LLaMA-3-8B** | | | | | | | | | | | | |
| SliceGPT | 316.1936 | 307.9020 | 231.0099 | 447.8662 | 488.1207 | 316.6689 | 746.2088 | 802.4747 | 530.2491 | 1182.3573 | 1214.1042 | 830.1348 |
| SLEB | 20.4018 | 38.1157 | 20.7231 | 33.6369 | 53.7517 | 30.1520 | 63.8317 | 115.5843 | 49.7122 | **126.9445** | 190.5663 | 90.0171 |
| Shortened-LLaMA | 20.7231 | 37.5247 | 20.7231 | 79.4398 | 105.2407 | 50.4950 | 5928.3428 | 9774.1849 | 2048.7805 | 15138.5538 | 46630.0285 | 2113.8157 |
| ShortGPT | 23.8522 | 41.2128 | 22.7599 | 84.5633 | 119.2533 | 61.8678 | 2549.7485 | 2714.1931 | 1364.7820 | 15138.5538 | 46630.0285 | 2113.8157 |
| Ours | **18.5761** | **34.7047** | **19.1658** | **25.3905** | **45.9763** | **24.9969** | **45.2635** | **70.1054** | **33.1155** | 304.5240 | 173.5126 | **65.8579** |

Table 2: Perplexity results of different pruning methods on WikiText2, PTB, and C4 for LLaMA-2-7B-hf, Vicuna-7B-v1.3 and Meta-LLaMA-3-8B.

## 4.3 INFERENCE PERFORMANCE

**Zero-shot Performance** We compare depth-wise pruning methods on LLaMA-2-7B-hf across eight zero-shot multiple-choice tasks in Tab. 3. Across all model sizes, our method consistently achieves the highest or near-highest average performance. For instance, when pruning the model to 5.5B parameters, our approach reaches 0.5227 average accuracy, outperforming SliceGPT (0.3865) and Shortened-LLaMA (0.5050). Similar trends hold for 4.9B and 4.3B models, where baselines degrade more sharply, particularly on reasoning-focused tasks such as ARC-challenge and WSC273. These results indicate that our method effectively preserves layers critical for downstream reasoning, complementing the generative capability retention observed in the language modeling experiments.

| Params | Method | PIQA | HeSw | ARC-e | ARC-c | OBQA | Race | WSC273 | LAMBADA | MMLU | Average |
|---|---|---|---|---|---|---|---|---|---|---|---|
| 6.7B | LLaMA-2-7B-hf | 0.7807 | 0.7602 | 0.7630 | 0.4625 | 0.4420 | 0.3962 | 0.8059 | 0.7388 | 0.4177 | 0.6186 |
| 6.1B | SliceGPT | 0.6676 | 0.5299 | 0.4663 | 0.3020 | 0.3140 | 0.3397 | 0.8022 | 0.3152 | 0.2500 | 0.4430 |
| | SLEB | 0.7644 | 0.7180 | 0.7138 | 0.3985 | 0.4080 | 0.3550 | 0.7839 | 0.6216 | 0.3084 | 0.5635 |
| | Shortened-LLaMA | 0.7497 | 0.7298 | 0.7201 | 0.4360 | 0.4040 | 0.3799 | 0.8278 | 0.6301 | 0.3572 | **0.5816** |
| | ShortGPT | 0.7573 | 0.7162 | 0.7104 | 0.4292 | 0.4040 | 0.3847 | 0.7692 | 0.6167 | 0.351 | 0.5709 |
| | Ours | 0.7709 | 0.7185 | 0.7231 | 0.4096 | 0.3940 | 0.3694 | 0.7912 | 0.6682 | 0.3663 | 0.5782 |
| 5.5B | SliceGPT | 0.6077 | 0.4270 | 0.3590 | 0.2756 | 0.2700 | 0.3081 | 0.7473 | 0.2395 | 0.2441 | 0.3865 |
| | SLEB | 0.7301 | 0.6656 | 0.6700 | 0.3951 | 0.3880 | 0.3502 | 0.7399 | 0.4477 | 0.2378 | 0.5138 |
| | Shortened-LLaMA | 0.7095 | 0.6528 | 0.5934 | 0.3797 | 0.3740 | 0.3330 | 0.7692 | 0.4747 | 0.2589 | 0.5050 |
| | ShortGPT | 0.7095 | 0.6528 | 0.5934 | 0.3797 | 0.3740 | 0.3330 | 0.7692 | 0.4747 | 0.2589 | 0.5050 |
| | Ours | 0.7372 | 0.6719 | 0.6473 | 0.3729 | 0.3860 | 0.3646 | 0.7546 | 0.4739 | 0.2955 | **0.5227** |
| 4.9B | SliceGPT | 0.5887 | 0.3856 | 0.3245 | 0.2619 | 0.2700 | 0.2947 | 0.7253 | 0.1846 | 0.2452 | 0.3645 |
| | SLEB | 0.6910 | 0.5640 | 0.5947 | 0.3251 | 0.3520 | 0.3263 | 0.6850 | 0.3134 | 0.2372 | 0.4543 |
| | Shortened-LLaMA | 0.6485 | 0.5617 | 0.4802 | 0.3276 | 0.3280 | 0.3225 | 0.7143 | 0.2915 | 0.3811 | 0.4506 |
| | ShortGPT | 0.6485 | 0.5617 | 0.4802 | 0.3276 | 0.3280 | 0.3225 | 0.7143 | 0.2915 | 0.3811 | 0.4506 |
| | Ours | 0.7046 | 0.5840 | 0.5821 | 0.3404 | 0.3600 | 0.3301 | 0.7033 | 0.3427 | 0.2725 | **0.4689** |
| 4.3B | SliceGPT | 0.5718 | 0.3475 | 0.3077 | 0.2500 | 0.2560 | 0.2775 | 0.6923 | 0.1450 | 0.2493 | 0.3441 |
| | SLEB | 0.6186 | 0.4665 | 0.4491 | 0.3020 | 0.3100 | 0.3024 | 0.5788 | 0.1527 | 0.2507 | 0.3812 |
| | Shortened-LLaMA | 0.6023 | 0.4430 | 0.3611 | 0.3063 | 0.2760 | 0.2909 | 0.6081 | 0.0505 | 0.3373 | 0.3640 |
| | ShortGPT | 0.5952 | 0.4371 | 0.4158 | 0.3003 | 0.3500 | 0.2900 | 0.6886 | 0.1100 | 0.3332 | 0.3911 |
| | Ours | 0.6659 | 0.4618 | 0.4937 | 0.2876 | 0.3260 | 0.2989 | 0.6117 | 0.1731 | 0.2370 | **0.3951** |

Table 3: Performance comparison of models with different retained parameters using various pruning methods on eight zero-shot benchmark datasets. Higher values indicate better performance.

**Adversarial Reasoning Robustness** To evaluate robustness under adversarial reasoning, we test pruned models on the ANLI dataset (Nie et al., 2020), which consists of three increasingly difficult rounds (R1–R3). As shown in Fig. 3, our method achieves the highest average accuracy across all rounds and consistently surpasses baselines, ranking first on R1 (36.7%) and R3 (36.6%). These results indicate that by explicitly capturing inter-layer dependencies, our method better preserves adversarial reasoning robustness, highlighting its potential for deployment in scenarios requiring resilience to distribution shifts and adversarial perturbations.

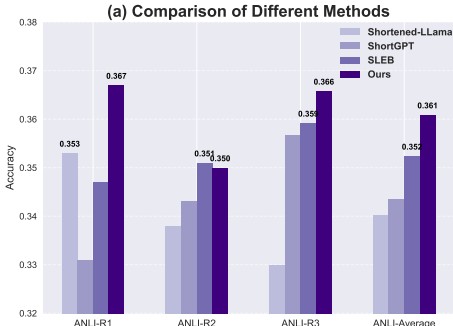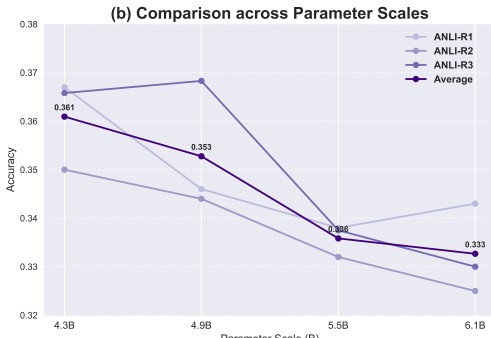

Figure 3: Adversarial reasoning accuracy on ANLI. (a) Comparison of pruned models on R1–R3 and average; only top two values per x are labeled. (b) Accuracy across parameter scales (3–12 layers, 6.1B–4.3B); only the average is labeled.

**Performance on larger-scale models**  To further demonstrate the generality of our method, we extend experiments to larger-scale models. Tab. 4 reports average zero-shot accuracy across eight tasks for Meta-LLaMA-3-8B and LLaMA-2-13B-hf. Our approach consistently outperforms or matches depth-wise baselines across pruning levels. For instance, at 9.2B parameters on LLaMA-2-13B-hf, our method achieves 0.5327 accuracy, compared to 0.3950 for SliceGPT and 0.4825 for Shortened-LLaMA, demonstrating robust generalization at high pruning ratios.

| (a) Meta-LLaMA-3-8B | | | | (b) LLaMA-2-13B-hf | | | |
|---|---|---|---|---|---|---|---|
| Method | 7.4B | 6.1B | 5.4B | Method | 11.8B | 10.5B | 9.2B |
| SliceGPT | 0.4465 | 0.3620 | 0.3296 | SliceGPT | 0.4959 | 0.4286 | 0.3950 |
| SLEB | 0.6109 | 0.4526 | 0.3696 | SLEB | 0.6393 | 0.5762 | 0.5289 |
| Shortened-LLaMA | 0.6299 | 0.3323 | 0.3576 | Shortened-LLaMA | 0.6349 | 0.5340 | 0.4825 |
| ShortGPT | 0.6054 | 0.3598 | 0.3576 | ShortGPT | 0.6456 | 0.5878 | 0.4825 |
| Ours | **0.6354** | **0.4676** | **0.3912** | Ours | **0.6470** | **0.5970** | **0.5327** |

Table 4: Average zero-shot accuracy on eight datasets for Meta-LLaMA-3-8B (a) and LLaMA-2-13B-hf (b) with different parameter sizes. Higher values indicate better performance.

## 4.4 Comparison with Width-wise pruning method

We extend our analysis to width-wise pruning methods on LLaMA-2-7B-hf, as shown in Fig. 4. Using the PTB dataset as a representative case, our method consistently achieves lower perplexity than width-wise pruning across different sparsity levels, confirming its superior ability to preserve generative capacity. For example, at 4.3B parameters, our model achieves a PPL of 105.2, substantially outperforming Wanda-sp and LLM-Pruner. In terms of efficiency, depth-wise pruning proves more favorable: removing layers leads to consistent improvements in both throughput and latency as pruning ratios increase, while width-wise pruning exhibits limited gains. These improvements are realized without additional memory overhead, with usage remaining in the range of 8.3–11.8 GB. Overall, our method outperforms width-wise pruning both in preserving language modeling quality and in delivering scalable runtime efficiency.

## 4.5 performance in non-transformer LLM

We apply our pruning strategy to RWKV-4-World-7B and Mamba-2.8B, and evaluate perplexity on WikiText2, PTB, and C4 under varying pruning scales in Tab. 5. Despite progressive layer reduction, our method maintains the overall generative performance of these non-Transformer models. For example, RWKV-7B retains a PPL of 56.3313 at 5.6B parameters before degradation becomes significant at more aggressive pruning. Similar robustness is observed on Mamba-2.8B, indicating that our pruning method generalizes beyond Transformer architectures and effectively preserves language modeling quality across diverse backbones.

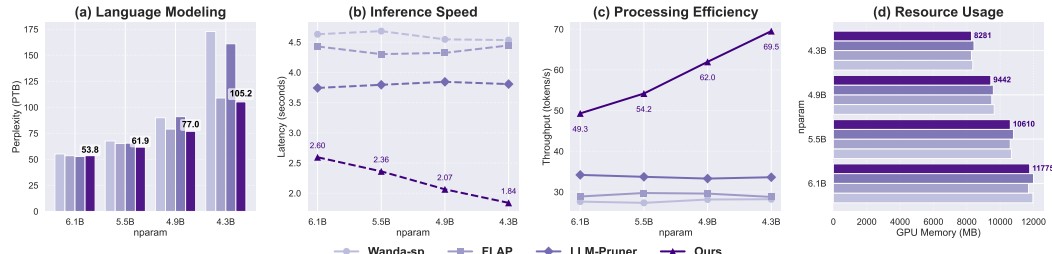

Figure 4: Comparison with structured width pruning methods (Wanda-sp, FLAP, LLM-Pruner) on PTB and system efficiency metrics across progressively reduced parameter budgets. Our method consistently achieves the best trade-off between perplexity, latency, throughput, and GPU memory.

| | | (a) RWKV-4-World-7B | | | | | (b) Mamba-2.8B | | |
|---|---|---|---|---|---|---|---|---|---|
| Params | Method | PPL_WikiText2 | PPL_PTB | PPL_C4 | Params | Method | PPL_WikiText2 | PPL_PTB | PPL_C4 |
| 6.2B | ShortGPT | 38.7159 | 61.8678 | 31.5990 | 2.5B | ShortGPT | 378.9863 | 1865.4358 | 391.0166 |
| | Ours | 34.1666 | 65.8579 | 32.0966 | | Ours | 24.2278 | 43.1907 | 22.0596 |
| 5.6B | ShortGPT | 90.0171 | 179.0204 | 67.9485 | 2.3B | ShortGPT | 4074.4865 | 15138.5538 | 4074.4865 |
| | Ours | 56.3313 | 105.2407 | 48.9415 | | Ours | 31.1091 | 53.7517 | 26.1965 |
| 4.9B | ShortGPT | 252.4593 | 471.6560 | 179.0204 | 2.0B | ShortGPT | 98715.7710 | 143630.5993 | 49637.4069 |
| | Ours | 130.9742 | 252.4593 | 95.8227 | | Ours | 41.2128 | 72.3308 | 33.6369 |

Table 5: PPL on WikiText2, PTB and C4 for non-Transformer models RWKV-4-World-7B (a) and Mamba-2.8B (b) with different parameter sizes. Lower values indicate better performance.

## 4.6 COMPATIBILITY WITH POST-TRAINING QUANTIZATION

Post-training quantization (PTQ) is a common technique to reduce memory usage during LLM inference. We evaluate its compatibility with our method by applying GPTQ (Frantar et al., 2023) to LLaMA-2-7B-hf at different pruning scales. As illustrated in Fig. 5, pruning and quantization demonstrate strong compatibility: quantization incurs only a modest increase in perplexity, while their combination effectively improves throughput and further amplifies memory savings—for example, reducing usage from 12.9 GB to 4.8 GB on the 6.7B variant. We further consider the influence by different integration orders of our pruning strategy and 4-bit quantization in Tab. 6. The results show that the language modeling performance differences across orders are generally small, confirming that our method is highly compatible with PTQ. Interestingly, we observe that placing the pruning step last often achieves the lowest perplexity. We attribute this to our method's explicit consideration of inter-layer dependencies: by analyzing the layer contributions after quantization, pruning decisions can be made on a representation that is closer to the model's final inference form, thereby yielding better retention of critical capacity.

| Scheme | PPL_WikiText2 | PPL_PTB | PPL_C4 |
|---|---|---|---|
| Remove 3 layers by ours then 4-bit quantization | 15.4001 | 56.3313 | 13.3799 |
| 4-bit quantization then remove 3 layers by ours | **14.9263** | 56.3313 | **13.1724** |
| Remove 6 layers by ours then 4-bit quantization | 19.7741 | 65.8579 | 16.9137 |
| Remove 3 layers by ours then 4-bit quantization then remove 3 layers by ours | 19.4676 | 67.9485 | 16.3933 |
| 4-bit quantization then remove 6 layers by ours | **18.5761** | 65.8579 | **15.8890** |

Table 6: Perplexity results on LLaMA-2-7B-hf under different integration orders of our pruning strategy and 4-bit quantization.

## 5 CONCLUSION

We formulate model compression as a cooperative game among layers, enabling principled estimation of inter-layer dependencies via a lightweight surrogate model. Extensive benchmarks show that our approach consistently outperforms depth-wise and width-wise pruning baselines, achiev-

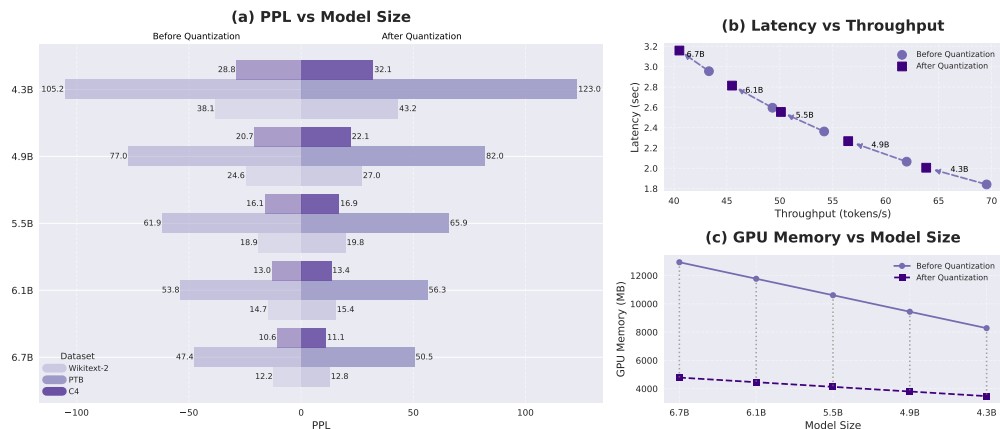

Figure 5: Evaluation before and after quantization across model sizes: (a) PPL (left / right bars), (b) Latency vs Throughput (circles / squares), and (c) GPU Memory (solid / dashed lines).

ing lower perplexity and higher zero-shot accuracy while delivering superior efficiency in latency, throughput, and memory usage. Evaluations on larger-scale models and non-Transformer architectures further underscore the generality of our method. Moreover, compatibility with post-training quantization highlights the potential for practical deployment. By introducing a game-theoretic perspective to model compression, our approach provides a novel framework for systematically and efficiently pruning large language models, achieving a balance between performance preservation and practical efficiency.

**Ethics Statement** This work adheres to the ICLR Code of Ethics. Our study exclusively uses publicly available datasets (e.g., WikiText2, PTB, C4, BookCorpus) that do not contain personally identifiable information. We believe that our work, which focuses on efficient pruning, primarily contributes to reducing the computational cost of training and deploying large models. We have disclosed all relevant details and ensured research integrity in accordance with the Code of Ethics.

**Reproducibility Statement** We have taken multiple steps to ensure reproducibility. All experimental settings, including datasets, hyperparameters, model configurations, and evaluation metrics, are detailed in the main text and Appendix. Algorithmic details and ablation studies are also included. To further facilitate reproduction, we provide an anonymized code repository in the supplementary material, containing scripts for data preprocessing, model pruning, training, and evaluation. These resources allow others to reproduce our results and verify the claims made in this paper.

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

## A    USAGE OF LARGE LANGUAGE MODELS

We employed a large language model to assist in polishing the language of this paper. Its use was restricted to improving linguistic fluency and reducing grammatical inaccuracies, with the goal of providing a clearer and more accessible reading experience. All research ideas, experimental designs, and conclusions were conceived and validated solely by the authors.

## B    METHOD DETAILS

### B.1    SURROGATE MODEL

As shown in Fig. 6, we propose using a lightweight surrogate network to estimate layer-wise marginal contributions. The surrogate network takes binary mask as input, projects it into a hidden dimension twice as large (input dimension = number of layers, hidden width = 2 × input), applies a CELU activation, and finally outputs a single scalar through a sigmoid activation.

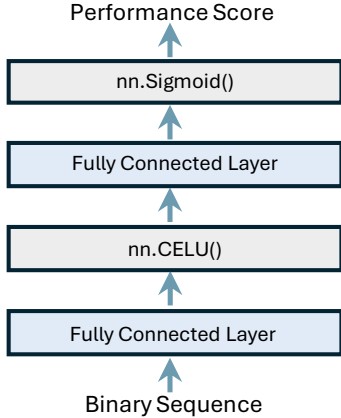

Figure 6: Framework of our surrogate network.

We use stochastic gradient descent with momentum 0.9, an initial learning rate of 0.008, and a step-decay scheduler that reduces the learning rate by a factor of 0.1 every 100 epochs in the training process. The model is optimized with mean squared error loss, trained with a batch size of 300, and produces both the learned parameters and the training loss curve as outputs (see Tab. 7).

| Setting | Value | Setting | Value |
|---|---|---|---|
| Input Dim | 32 | Hidden Dim | 64 (CELU) |
| Output | 1 (Sigmoid) | Optimizer | SGD + Momentum 0.9 |
| Learning Rate | 0.008 (decay ×0.1 every 100 epochs) | Loss | MSE |

Table 7: Summary of hyperparameters used for our surrogate network.

We tested the surrogate model's performance using the LLaMA-2-7B-hf model as an example (see Tab.8). By calculating error metrics and $R^2$ values, we found that the surrogate model performs stably when changing the random seed or the number of test samples. For different mask configurations, the model maintains high predictive accuracy when the test mask closely matches the training mask. However, if the test mask differs significantly from the training mask, the model's predictive ability declines, which is expected. In practical experiments, the training mask is adjusted according to the desired pruning ratio, meaning that the test mask is similar to the training mask. This ensures the usefulness of the surrogate model's predictions.

| Case | Test Samples | Seed | Test Mask | $R^2$ |
|---|---|---|---|---|
| Same Seed, Same Mask | 200 | 42 | ks = (30, 27, 24, 21, 18) | 0.9360 |
| | 500 | 42 | ks = (30, 27, 24, 21, 18) | 0.9492 |
| | 1000 | 42 | ks = (30, 27, 24, 21, 18) | 0.9464 |
| Different Seeds, Same Mask | 500 | 500 | ks = (30, 27, 24, 21, 18) | 0.9359 |
| | 500 | 1234 | ks = (30, 27, 24, 21, 18) | 0.9402 |
| | 500 | 99999 | ks = (30, 27, 24, 21, 18) | 0.9400 |
| Same Seed, Different Masks | 500 | 42 | ks = (16, 27, 24, 21, 18) | 0.9230 |
| | 500 | 42 | ks = (16, 14, 24, 21, 18) | 0.8658 |
| | 500 | 42 | ks = (16, 14, 12, 21, 18) | 0.2466 |

Table 8: Surrogate Model Performance on LLaMA-2-7B-hf Model.

## B.2 HYPERPARAMETERS SETTING

We use BookCorpus as the calibration dataset, sampling 10 prompts of up to 128 tokens, and compute baseline perplexity as the normalization reference for evaluating masked models. All models are run in float16 precision with a fixed random seed of 42. The pruning pipeline follows three stages: Step 1 evaluates 8,000 randomly masked sub-networks, Step 2 trains the surrogate for 200 epochs, and Step 3 scores 80,000 masks via the surrogate. Batch sizes are set to 45 for mask evaluation and 300 for surrogate training. While in principle different architectures may adopt distinct configurations, our experiments follow the unified settings summarized in Tab. 9. The only variation lies in the predefined Hamming weight sets, which are scaled to match model depth, e.g., $\{30, 27, 24, 21, 18\}$ for 32-layer models, $\{36, 32, 28, 24, 20\}$ for 40-layer models, and $\{60, 56, 52, 48, 42, 38, 34\}$ for 64-layer models. This design ensures fair comparison across models while preserving flexibility for adaptation to other architectures.

| Component | Setting | Value | Setting | Value |
|---|---|---|---|---|
| | Dataset | BookCorpus (10 samples, 128 tokens) | Random Seed | 42 |
| | Batch size of Mask Evaluation | 45 | Batch size of training | 300 |
| LLaMA-2-7B-hf | Number of Layers | 32 | Precision | float16 |
| | Hamming Weights | {30, 27, 24, 21, 18} | Step 1 Samples | 8,000 |
| | Step 2 Epochs | 200 | Step 3 Samples | 80,000 |
| Vicuna-7B-v1.3 | Number of Layers | 32 | Precision | float16 |
| | Hamming Weights | {30, 27, 24, 21, 18} | Step 1 Samples | 8,000 |
| | Step 2 Epochs | 200 | Step 3 Samples | 80,000 |
| Meta-LLaMA-3-8B | Number of Layers | 32 | Precision | float16 |
| | Hamming Weights | {30, 27, 24, 21, 18} | Step 1 Samples | 8,000 |
| | Step 2 Epochs | 200 | Step 3 Samples | 80,000 |
| LLaMA-2-13B-hf | Number of Layers | 40 | Precision | float16 |
| | Hamming Weights | {36, 32, 28, 24, 20} | Step 1 Samples | 8,000 |
| | Step 2 Epochs | 200 | Step 3 Samples | 80,000 |
| RWKV-4-World-7B | Number of Layers | 32 | Precision | float16 |
| | Hamming Weights | {30, 27, 24, 21} | Step 1 Samples | 12,000 |
| | Step 2 Epochs | 200 | Step 3 Samples | 80,000 |
| Mamba-2.8B | Number of Layers | 64 | Precision | float16 |
| | Hamming Weights | {60, 56, 52, 48, 42, 38, 34} | Step 1 Samples | 8,000 |
| | Step 2 Epochs | 200 | Step 3 Samples | 80,000 |

Table 9: Summary of hyperparameters used in our experiments.

### B.3 ALGORITHM PSEUDOCODE

Algorithm 1 provides the main procedure of our method, where layer contributions are estimated through mask perturbation, surrogate training, and Monte Carlo approximation. To support this process, Algorithm 2 describes the auxiliary strategy used to generate binary masks with predefined Hamming weights, ensuring sufficient diversity of pruning patterns for stable surrogate learning. Together, these components form the backbone of our pruning framework.

---

**Algorithm 1** Layer Contribution Estimation via Mask Perturbation and Surrogate Learning

---

**Require:** Pretrained model $\mathcal{M}$ with $L$ layers; validation set $\mathcal{D}$; number of masks $N$; Monte Carlo samples $M$.
**Ensure:** Estimated layer contribution scores $C \in \mathbb{R}^L$.
 1: **Stage 1: Data Generation (Mask Evaluation)**
 2: **for** each mask $\mathbf{m}_j$ sampled by stratified Hamming weight **do**
 3:     Apply mask $\mathbf{m}_j$ to $\mathcal{M}$ and compute masked model perplexity $\text{PPL}_{\text{masked}}(\mathbf{m}_j)$
 4:     Compute performance score:
$$s(\mathbf{m}_j) = \frac{\text{PPL}_{\text{baseline}}}{\text{PPL}_{\text{masked}}(\mathbf{m}_j)}$$
 5: **end for**
 6: **Stage 2: Surrogate Training and Layer Contribution Estimation**
 7: Train surrogate network $f_\theta$ to predict mask scores with MSE loss:
$$\mathcal{L}(\theta) = \frac{1}{N} \sum_{j=1}^{N} \left( f_\theta(\mathbf{m}_j) - s(\mathbf{m}_j) \right)^2$$
 8: **for** $\ell = 1$ to $L$ **do**                              ▷ Estimate contribution for each layer
 9:     $A \leftarrow 0$
10:     **for** $m = 1$ to $M$ **do**                 ▷ Monte Carlo approximation over random masks
11:         Sample a base mask $\mathbf{m}$
12:         Compute marginal contribution:
$$\Delta = f_\theta(\mathbf{m}_{+\ell}) - f_\theta(\mathbf{m}_{-\ell})$$
13:         $A \leftarrow A + \Delta$
14:     **end for**
15:     $C_\ell \leftarrow A/M$
16: **end for**
17: **return** $C = (C_1, C_2, \ldots, C_L)$

---

## C EXPERIMENTAL SETTING

### C.1 BASELINE METHODS

We primarily consider the width-wise pruning methods and depth-wise pruning methods as our baseline methods in our analysis. The specific information of baseline methods are described below, where we use their official code for implementation. To ensure a fair comparison, we employ the same calibration dataset across all methods.

#### C.1.1 WIDTH-WISE METHOD

The width-wise pruning baselines considered include Wanda-sp, FLAP, and LLM-Pruner. Wanda-sp is a structured variant of Wanda (Sun et al., 2024b), where the original metric—based on the product of weight magnitudes and input activation norms—is extended to exploit shared dimensions across modules (An et al., 2024). FLAP (An et al., 2024) is a retraining-free structured pruning framework that evaluates the recoverability of feature maps via the fluctuation pruning index. It adaptively determines the compressed structure using normalized importance scores and introduces

---

**Algorithm 2** Stratified Mask Sampling by Hamming Weight

---

**Require:** Number of layers $L$, total samples $M$, predefined Hamming weights $K$
**Ensure:** A set of binary masks $\mathcal{M}$
 1: **if** per_k is not given **then**
 2:     $q \leftarrow \lfloor M/|K| \rfloor, r \leftarrow M \bmod |K|$
 3:     per_k $\leftarrow [q + 1$ for first $r$ values, $q$ for others]
 4: **end if**
 5: $\mathcal{M} \leftarrow \emptyset$
 6: **for** each $(k, c)$ in $(K, \text{per\_k})$ **do**
 7:     **for** $i = 1$ to $c$ **do**
 8:         idx $\leftarrow$ randomly select $k$ distinct indices from $\{1, \ldots, L\}$
 9:         $m \leftarrow$ binary vector of length $L$ with $m[\text{idx}] = 1$, others $= 0$
10:         $\mathcal{M} \leftarrow \mathcal{M} \cup \{m\}$
11:     **end for**
12: **end for**
13: **return** $\mathcal{M}$

---

bias correction to pruned feature maps to mitigate accuracy loss. As in the original paper, we adopt the default configuration: pruning metric = WIFV (among [IFV, WIFV, WIFN, N/A]) and global structure = AL-AM (among [UL-UM, UL-MM, AL-MM, AL-AM]). LLM-Pruner (Ma et al., 2023) employs a Taylor-based importance metric to prune attention heads in MHA and neurons in FFN, operating locally within modules while preserving dimension consistency across blocks; we follow the original setting of keeping the first four and last two layers intact. The specific pruning ratios applied to LLaMA-2-7B-hf are detailed in Tab. 10.

| Method | Remove 3 layers | | Remove 6 layers | | Remove 9 layers | | Remove 12 layers | |
|---|---|---|---|---|---|---|---|---|
| | Pruned Ratio | Params | Pruned Ratio | Params | Pruned Ratio | Params | Pruned Ratio | Params |
| Wanda-sp | 0.1 | 6104813568 | 0.19 | 5513809920 | 0.29 | 4879552512 | 0.38 | 4288548864 |
| FLAP | 0.1 | 6091898880 | 0.19 | 5509578752 | 0.28 | 4924891136 | 0.38 | 4279099392 |
| LLM_Pruner | 0.12 | 6152794112 | 0.24 | 5512646656 | 0.35 | 4907642880 | 0.46 | 4357165056 |

Table 10: Pruned ratio settings of width-wise method on LLaMA-2-7B-hf.

### C.1.2 DEPTH-WISE METHOD

Depth pruning methods adopt the Transformer block as the basic pruning unit. We evaluate four representative approaches: SliceGPT, SLEB, ShortGPT, and Shortened-LLaMA. SliceGPT (Ashkboos et al., 2024) is a post-training sparsification technique that replaces each weight matrix with a smaller dense matrix, thereby reducing the embedding dimension; the pruning ratios used in our experiments are summarized in Tab. 11. SLEB (Song et al., 2024) employs a logit-based criterion to identify redundant blocks and iteratively updates importance scores after block removal. Although designed for a no-retraining scenario, SLEB suffers from noticeable performance degradation at higher pruning rates. ShortGPT (Men et al., 2024) introduces the Block Influence (BI) metric, which quantifies the contribution of each block by measuring the similarity between its input and output representations. Shortened-LLaMA (Kim et al., 2024) determines block importance based on perplexity (PPL) sensitivity and removes low-importance layers accordingly. The specific pruned layer indices for SLEB, ShortGPT, and Shortened-LLaMA across four different model architectures are provided in Tab. 12, Tab. 13, Tab. 14, and Tab. 15.

| Method | Remove 3 layers | | Remove 6 layers | | Remove 9 layers | | Remove 12 layers | |
|---|---|---|---|---|---|---|---|---|
| | Pruned Ratio | Params | Pruned Ratio | Params | Pruned Ratio | Params | Pruned Ratio | Params |
| LLaMA-2-7B-hf | 0.2 | 6105940928 | 0.3 | 5292914432 | 0.35 | 4886502400 | 0.4 | 4500862400 |
| Vicuna-7b-v1.3 | 0.2 | 6105940928 | 0.3 | 5292914432 | 0.35 | 4886502400 | 0.4 | 4500862400 |
| Meta-Llama-3-8B | 0.19 | 7309430528 | 0.25 | 6775635968 | 0.33 | 6057853888 | 0.33 | 6057853888 |

Table 11: Pruned ratio settings of SliceGPT.

| | Params | Method | Remove Layers Index |
|---|---|---|---|
| Remove 3 layers | 6.1B (6131265536) | SLEB | [15, 14, 24] |
| | | Shortened_llama | [28, 27, 25] |
| | | ShortGPT | [25, 27, 26] |
| | | Ours | [21, 23, 11] |
| Remove 6 layers | 5.5B (5524115456) | SLEB | [15, 14, 24, 13, 25, 22] |
| | | Shortened_llama | [28, 27, 25, 29, 26, 24] |
| | | ShortGPT | [25, 27, 26, 24, 28, 29] |
| | | Ours | [21, 23, 11, 12, 18, 24] |
| Remove 9 layers | 4.9B (4916965376) | SLEB | [15, 14, 24, 13, 25, 22, 8, 23, 12] |
| | | Shortened_llama | [28, 27, 25, 29, 26, 24, 23, 21, 22] |
| | | ShortGPT | [25, 27, 26, 24, 28, 29, 23, 22, 21] |
| | | Ours | [21, 23, 11, 12, 18, 24, 10, 27, 25] |
| Remove 12 layers | 4.3B (4309815296) | SLEB | [15, 14, 24, 13, 25, 22, 8, 23, 12, 29, 21, 7] |
| | | Shortened_llama | [28, 27, 25, 29, 26, 24, 23, 21, 22, 20, 19, 18] |
| | | ShortGPT | [25, 27, 26, 24, 28, 29, 23, 22, 21, 19, 30, 20] |
| | | Ours | [21, 23, 11, 12, 18, 24, 10, 27, 25, 14, 8, 9] |

Table 12: Pruned layer index of depth-wise method on LLaMA-2-7B-hf.

| | Params | Method | Remove Layers Index |
|---|---|---|---|
| Remove 3 layers | 6.1B (6131265536) | SLEB | [7, 27, 24] |
| | | Shortened_llama | [27, 26, 24] |
| | | ShortGPT | [27, 28, 26] |
| | | Ours | [26, 24, 29] |
| Remove 6 layers | 5.5B (5524115456) | SLEB | [7, 27, 24, 17, 22, 10] |
| | | Shortened_llama | [27, 26, 24, 25, 29, 28] |
| | | ShortGPT | [27, 28, 26, 29, 25, 24] |
| | | Ours | [26, 24, 29, 27, 11, 10] |
| Remove 9 layers | 4.9B (4916965376) | SLEB | [7, 27, 24, 17, 22, 10, 26, 13, 14] |
| | | Shortened_llama | [27, 26, 24, 25, 29, 28, 23, 21, 22] |
| | | ShortGPT | [27, 28, 26, 29, 25, 24, 23, 22, 21] |
| | | Ours | [26, 27, 24, 29, 11, 12, 10, 22, 25] |
| Remove 12 layers | 4.3B (4309815296) | SLEB | [7, 27, 24, 17, 22, 10, 26, 13, 14, 8, 9, 25] |
| | | Shortened_llama | [27, 26, 24, 25, 29, 28, 23, 21, 22, 20, 19, 18] |
| | | ShortGPT | [27, 28, 26, 29, 25, 24, 23, 22, 21, 30, 20, 19] |
| | | Ours | [26,24,29,27,11,10,25,12,22,9,20,8] |

Table 13: Pruned layer index of depth-wise method on Vicuna-7B-v1.3.

## C.2 SELECTED LAYERS OF NON-TRANSFORMER MODELS

We evaluate our method on non-Transformer architectures, including RWKV-4-World-7B and Mamba-2.8B. Tab. 16 reports the specific layer indices removed by our approach.

## C.3 SELECTED LAYERS OF QUANTIZED MODEL

Our pruning method can be combined with quantization to further decrease memory usage. To validate this aspect, we apply 4-bit GPTQ to our pruned models, using 128 randomly sampled sequences with 2048 tokens from the C4 dataset as calibration data. The specific layer indices removed under different integration orders of pruning and quantization are summarized in Tab. 17.

## D DETAILED RESULTS OF ZERO-SHOT EVALUATION

In the main text, we reported the average performance of our method on eight zero-shot tasks for different model architectures. To provide a more comprehensive view, we include the detailed per-task results in Tab. 18 and Tab. 19.

| | Params | Method | Remove Layers Index |
|---|---|---|---|
| Remove 3 layers | 7.4B (7375925248) | SLEB | [7, 25, 18] |
| | | Shortened_llama | [24, 25, 26] |
| | | ShortGPT | [25, 27, 26] |
| | | Ours | [8, 25, 26] |
| Remove 6 layers | 6.7B (6721589248) | SLEB | [7, 25, 18, 23, 28, 26] |
| | | Shortened_llama | [24, 25, 26, 28, 29, 23] |
| | | ShortGPT | [25, 27, 26, 24, 28, 23] |
| | | Ours | [8, 25, 26, 10, 11, 19] |
| Remove 9 layers | 6.1B (6067253248) | SLEB | [7, 25, 18, 23, 28, 26, 14, 13, 22] |
| | | Shortened_llama | [24, 25, 26, 28, 29, 23, 27, 22, 20] |
| | | ShortGPT | [25, 27, 26, 24, 28, 23, 22, 29, 21] |
| | | Ours | [8, 25, 26, 10, 11, 19, 24, 9, 12] |
| Remove 12 layers | 5.4B (5412917248) | SLEB | [7, 25, 18, 23, 28, 26, 14, 13, 22, 10, 8, 21] |
| | | Shortened_llama | [24, 25, 26, 28, 29, 23, 27, 22, 20, 19, 21, 18] |
| | | ShortGPT | [25, 27, 26, 24, 28, 23, 22, 29, 21, 20, 19, 18] |
| | | Ours | [8, 25, 26, 10, 11, 19, 24, 9, 12, 23, 21, 22] |

Table 14: Pruned layer index of depth-wise method on Meta-LLaMA-3-8B.

| | Params | Method | Remove Layers Index |
|---|---|---|---|
| Remove 4 layers | 11.7B (11747046400) | SLEB | [15, 28, 31, 29] |
| | | Shortened_llama | [35, 33, 34, 36] |
| | | ShortGPT | [33, 31, 32, 30] |
| | | Ours | [31, 27, 26, 29] |
| Remove 8 layers | 10.5B (10478228480) | SLEB | [15, 28, 31, 29, 22, 13, 18, 8] |
| | | Shortened_llama | [35, 33, 34, 36, 37, 31, 32, 30] |
| | | ShortGPT | [33, 31, 32, 30, 34, 35, 29, 28] |
| | | Ours | [31, 27, 26, 29, 25, 28, 10, 24] |
| Remove 12 layers | 9.2B (9209410560) | SLEB | [15, 28, 31, 29, 22, 13, 18, 8, 30, 27, 19, 33] |
| | | Shortened_llama | [35, 33, 34, 36, 37, 31, 32, 30, 28, 27, 29, 26] |
| | | ShortGPT | [33, 31, 32, 30, 34, 35, 29, 28, 27, 36, 37, 26] |
| | | Ours | [31, 27, 26, 29, 25, 28, 10, 24, 15, 22, 23, 30] |

Table 15: Pruned layer index of depth-wise method on LLaMA-2-13B-hf.

# E  FURTHER RESULTS OF LoRA RETRAINING

LoRA provides an efficient approach to refining large language models (LLMs) with significantly reduced computational overhead. In our experiments, we follow the setup in Ma et al. (2023) and insert LoRA adapters into every projection weight matrix of the Transformer blocks. Specifically, we adopt a LoRA rank of 8, train with a learning rate of 1e-4, a batch size of 64, and run for 2 epochs. The entire fine-tuning process is lightweight: it requires only a single GPU and incurs negligible retraining cost compared to full model fine-tuning.

As shown in Fig. 7, LoRA fine-tuning consistently lowers perplexity across different pruning ratios, and the benefit becomes more pronounced as pruning intensifies. When compared at the 4.3B scale, our method not only maintains the lowest perplexity prior to fine-tuning but also preserves this advantage after LoRA is applied, underscoring both its robustness and its compatibility with lightweight adaptation strategies.

# F  ABLATION STUDY

## F.1  CALIBRATION DATASET

We analyze the effect of calibration settings on pruning robustness in Table 20. With mild pruning, different datasets lead to comparable outcomes, but discrepancies become evident under more aggressive pruning: WikiText2 calibration degrades PTB perplexity more severely, while C4 shows less stability on its own domain. Increasing the number of calibration samples (from 10 to 50 or 100) delays degradation in the moderate pruning, yet once more than ten layers are removed, all

| Model | | Params | Remove Layers Index |
|---|---|---|---|
| RWKV-4-World-7B | Remove 6 layers | 6.2B (6208757760) | [12, 20, 13, 29, 23, 5] |
| | Remove 9 layers | 5.6B (5554311168) | [12, 20, 13, 29, 23, 5, 17, 25, 15] |
| | Remove 12 layers | 4.9B (4899864576) | [12, 20, 13, 29, 23, 5, 17, 25, 15, 22, 18, 3] |
| Mamba-2.8B | Remove 6 layers | 2.5B (2520880640) | [8, 5, 3, 7, 4, 13] |
| | Remove 12 layers | 2.3B (2273415680) | [8, 5, 3, 7, 4, 13, 10, 9, 31, 6, 32, 21] |
| | Remove 18 layers | 2.0B (2025950720) | [8, 5, 3, 7, 4, 13, 10, 9, 31, 6, 32, 21, 14, 22, 12, 35, 34, 27] |

Table 16: Pruned layer index of our pruning method on non-Transformer model.

| Scheme | Pruned layer index |
|---|---|
| Remove 3 layers by ours then 4-bit quantization | [21, 23, 11] |
| 4-bit quantization then remove 3 layers by ours | [21, 11, 12] |
| Remove 6 layers by ours then 4-bit quantization | [21, 23, 11, 12, 18, 24] |
| Remove 3 layers by ours then 4-bit quantization then remove 3 layers by ours | [21, 23, 11] + [11, 10, 8] |
| 4-bit quantization then remove 6 layers by ours | [21, 11, 12, 25, 23, 10] |

Table 17: Perplexity results on LLaMA-2-7B-hf under different integration orders of our pruning strategy and 4-bit quantization.

settings deteriorate similarly. We attribute this to distributional biases across datasets, as well as the surrogate model's tendency to overfit noisy calibration signals when exposed to larger and more heterogeneous sets. These results indicate that pruning robustness depends little dataset choice, benefits only marginally from sample size, and is fundamentally limited by the depth of pruning.

## F.2 SIMULATION NUMBER FOR LAYER PRUNING

To examine the effect of the number of Monte Carlo simulations (Simu_Num) on pruning performance, we conduct an ablation study on LLaMA-2-7B-hf with simulation counts ranging from 3,000 to 15,000, as shown in Tab. 21. In the lightly pruned regime, all settings yield similar perplexity across WikiText2, PTB, and C4, indicating robustness to simulation count. As pruning deepens, schemes with larger Simu_Num achieve slightly lower perplexity, reflecting more accurate estimation of layer importance. For example, Scheme 9 (Simu_Num=15,000) consistently outperforms Scheme 7 (Simu_Num=3,000) when 12 layers are pruned. Although perplexity increases with pruning depth across all settings, the relative ranking of masks remains stable, suggesting that our method reliably identifies critical layers even with fewer simulations.

## F.3 HAMMING WEIGHT CONSTRAINT FOR MASK GENERATION

We analyze the effect of incorporating a Hamming Weight constraint in Step 1 during mask generation. Tab. 22 compares Scheme 1, which adopts stratified sampling over predefined Hamming Weights $ks = (30, 27, 24, 21, 18)$, with Scheme 13, which generates masks fully at random. The results show a clear advantage of using the Hamming Weight constraint. In the lightly pruned regime (first pruning step), both approaches achieve similar PPL, but as pruning proceeds, random mask generation in Scheme 10 quickly leads to sharp performance degradation. For instance, after pruning 9 layers, Scheme 1 yields PPLs of 24.6, 77.0, and 20.7 on WikiText2, PTB, and C4, respectively, while Scheme 10 under the same pruning depth reaches much higher values of 60.0, 190.6, and 43.9. The gap further widens when pruning 12 layers, where random sampling results in catastrophic degradation (PPL >100 on WikiText2).

To further explore the impact of the Hamming weight constraint, we tested several additional weight constraints (Schemes 10–12). The results show that different Hamming weight ranges lead to varying performance. In the case of light pruning (e.g., pruning 3 layers), the differences are minimal. However, as pruning depth increases, the effects become more pronounced. As we intervene more with the Hamming weights (from Scheme 10 to Scheme 12), the pruning performance progressively worsens, and this effect becomes more pronounced as the pruning depth increases.

The optimal performance of Scheme 1 further highlights that the choice of Hamming weight range should be tailored to the desired pruning ratio. A balanced Hamming weight range that matches

| Params | Method | PIQA | HeSw | ARC-e | ARC-c | OBQA | Race | WSC237 | LAMBADA |
|---|---|---|---|---|---|---|---|---|---|
| 7.4B | SliceGPT | 0.6371 | 0.5076 | 0.4512 | 0.2841 | 0.2980 | 0.3273 | 0.7436 | 0.3227 |
| | SLEB | 0.7726 | 0.6993 | 0.7643 | 0.4556 | 0.4160 | 0.3818 | 0.7949 | 0.6026 |
| | Shortened_llama | 0.7726 | 0.7569 | 0.7584 | 0.4761 | 0.4020 | 0.3837 | 0.8425 | 0.6472 |
| | ShortGPT | 0.7644 | 0.7566 | 0.7517 | 0.4863 | 0.4220 | 0.3990 | 0.8095 | 0.4535 |
| | Ours | 0.7797 | 0.7399 | 0.7466 | 0.4710 | 0.4240 | 0.3914 | 0.8315 | 0.6994 |
| 6.7B | SliceGPT | 0.6034 | 0.4440 | 0.3830 | 0.2585 | 0.2740 | 0.3110 | 0.7106 | 0.2827 |
| | SLEB | 0.7394 | 0.6352 | 0.6856 | 0.4215 | 0.3660 | 0.3770 | 0.7143 | 0.5447 |
| | Shortened_llama | 0.7155 | 0.6419 | 0.6557 | 0.4411 | 0.3760 | 0.3627 | 0.7399 | 0.3776 |
| | ShortGPT | 0.7247 | 0.6815 | 0.6263 | 0.4352 | 0.3700 | 0.3569 | 0.7399 | 0.3433 |
| | Ours | 0.7459 | 0.6473 | 0.6662 | 0.3840 | 0.3560 | 0.3292 | 0.7582 | 0.5581 |
| 6.1B | SliceGPT | 0.5773 | 0.3612 | 0.3249 | 0.2381 | 0.2540 | 0.2756 | 0.6593 | 0.2057 |
| | SLEB | 0.6768 | 0.5218 | 0.5362 | 0.3268 | 0.3280 | 0.2900 | 0.6484 | 0.2930 |
| | Shortened_llama | 0.5881 | 0.2912 | 0.3746 | 0.3038 | 0.2800 | 0.2536 | 0.5385 | 0.0287 |
| | ShortGPT | 0.6143 | 0.3159 | 0.3994 | 0.3183 | 0.3000 | 0.2440 | 0.6374 | 0.0487 |
| | Ours | 0.7138 | 0.5561 | 0.5934 | 0.3242 | 0.3140 | 0.2995 | 0.6447 | 0.2952 |
| 5.4B | SliceGPT | 0.5571 | 0.3106 | 0.3114 | 0.2312 | 0.2560 | 0.2584 | 0.5897 | 0.1225 |
| | SLEB | 0.6246 | 0.4063 | 0.4125 | 0.2884 | 0.2780 | 0.2766 | 0.5934 | 0.0768 |
| | Shortened_llama | 0.5876 | 0.3774 | 0.3737 | 0.3038 | 0.2800 | 0.2699 | 0.6337 | 0.0349 |
| | ShortGPT | 0.5876 | 0.3774 | 0.3737 | 0.3038 | 0.2800 | 0.2699 | 0.6337 | 0.0349 |
| | Ours | 0.6474 | 0.4530 | 0.4583 | 0.2807 | 0.2940 | 0.2842 | 0.5971 | 0.1149 |

Table 18: Detailed Zero-shot Downstream Task Performance of Meta-LLaMA-3-8B.

| Params | Method | PIQA | HeSw | ARC-e | ARC-c | OBQA | Race | WSC237 | LAMBADA |
|---|---|---|---|---|---|---|---|---|---|
| 11.7B | SLEB | 0.7709 | 0.7593 | 0.7567 | 0.4608 | 0.4360 | 0.4038 | 0.8352 | 0.6918 |
| | Shortened_llama | 0.7709 | 0.7694 | 0.7563 | 0.4727 | 0.4420 | 0.4048 | 0.8278 | 0.6350 |
| | ShortGPT | 0.7726 | 0.7662 | 0.7614 | 0.4770 | 0.4460 | 0.4029 | 0.8645 | 0.6738 |
| | SliceGPT | 0.6801 | 0.5654 | 0.5189 | 0.3328 | 0.3380 | 0.3435 | 0.8535 | 0.3350 |
| | Ours | 0.7731 | 0.7733 | 0.7567 | 0.4667 | 0.4280 | 0.3971 | 0.8535 | 0.7279 |
| 10.5B | SLEB | 0.7470 | 0.6851 | 0.6902 | 0.4070 | 0.3840 | 0.3636 | 0.7729 | 0.5595 |
| | Shortened_llama | 0.7291 | 0.6536 | 0.6246 | 0.3857 | 0.4320 | 0.3455 | 0.7949 | 0.3062 |
| | ShortGPT | 0.7399 | 0.7240 | 0.6852 | 0.4377 | 0.4120 | 0.3828 | 0.7985 | 0.5224 |
| | SliceGPT | 0.6219 | 0.4525 | 0.4015 | 0.2910 | 0.2960 | 0.3177 | 0.7875 | 0.2608 |
| | Ours | 0.7443 | 0.7257 | 0.6890 | 0.4155 | 0.3840 | 0.3694 | 0.8095 | 0.6389 |
| 9.2B | SLEB | 0.7193 | 0.6295 | 0.6132 | 0.3643 | 0.3580 | 0.3445 | 0.7289 | 0.4735 |
| | Shortened_llama | 0.6801 | 0.5793 | 0.5535 | 0.3575 | 0.3780 | 0.3062 | 0.7839 | 0.2212 |
| | ShortGPT | 0.6801 | 0.5793 | 0.5535 | 0.3575 | 0.3780 | 0.3062 | 0.7839 | 0.2212 |
| | SliceGPT | 0.5963 | 0.4096 | 0.3561 | 0.2790 | 0.2960 | 0.2852 | 0.7253 | 0.2123 |
| | Ours | 0.7013 | 0.6399 | 0.5930 | 0.3567 | 0.3740 | 0.3636 | 0.7289 | 0.5044 |
| 7.9B | SLEB | 0.6823 | 0.5485 | 0.5311 | 0.3328 | 0.3300 | 0.3081 | 0.6557 | 0.3569 |
| | Shortened_llama | 0.6230 | 0.4718 | 0.4524 | 0.3174 | 0.3560 | 0.2775 | 0.6630 | 0.1304 |
| | ShortGPT | 0.6230 | 0.4718 | 0.4524 | 0.3174 | 0.3560 | 0.2775 | 0.6630 | 0.1304 |
| | SliceGPT | 0.5642 | 0.3309 | 0.2976 | 0.2585 | 0.2500 | 0.2612 | 0.6520 | 0.1285 |
| | Ours | 0.6567 | 0.5265 | 0.4566 | 0.3046 | 0.3220 | 0.3263 | 0.6520 | 0.3419 |

Table 19: Detailed Zero-shot Downstream Task Performance of LLaMA-2-13B-hf.

the pruning requirements is essential to maintain performance while ensuring diverse and effective pruning patterns.

Overall, all constrained sampling schemes (Schemes 1, 10, 11, and 12) outperform random sampling (Scheme 13), underscoring the advantages of Hamming-weight-guided pruning. These findings suggest that stratified sampling by Hamming Weight stabilizes the Monte Carlo estimation of layer contributions, ensuring that sampled masks cover diverse pruning patterns in a more balanced manner. Without this constraint, random sampling can generate unbalanced or extreme masks, which may bias importance estimation and lead to suboptimal pruning. Ultimately, Hamming Weight-guided sampling significantly improves the robustness and effectiveness of our framework, particularly under deep pruning.

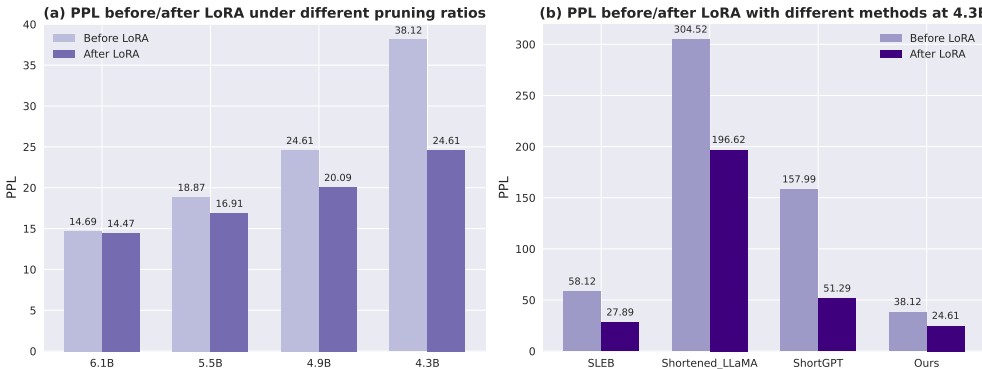

Figure 7: **Effect of LoRA fine-tuning on pruned models.** (a) PPL before and after LoRA fine-tuning under different pruning ratios using our method. (b) PPL before and after LoRA fine-tuning with different pruning methods at 4.3B parameters.

| Scheme | Calibration Dataset | Pruned Layer Index | PPL_WikiText2 | PPL_PTB | PPL_C4 | Other Setting |
|---|---|---|---|---|---|---|
| Scheme1 | Bookcorpus, num=10 | [21, 23, 11] | 14.6949 | 53.7517 | 12.9682 | |
| | | [21, 23, 11, 12, 18, 24] | 18.8686 | 61.8678 | 16.1392 | |
| | | [21, 23, 11, 12, 18, 24, 10, 27, 25] | 24.6093 | 76.9957 | 20.7231 | |
| | | [21, 23, 11, 12, 18, 24, 10, 27, 25, 14, 8, 9] | 38.1157 | 105.2407 | 28.7712 | |
| Scheme2 | C4, num=10 | [11, 21, 24] | 14.4671 | 53.7517 | 12.7671 | |
| | | [11, 21, 24, 14, 25, 23] | 18.8686 | 61.8678 | 16.1392 | |
| | | [11, 21, 24, 14, 25, 23, 9, 12, 8] | 24.9969 | 79.4398 | 20.4018 | |
| | | [11, 21, 24, 14, 25, 23, 9, 12, 8, 13, 27, 10] | 38.1157 | 148.4132 | 26.1965 | |
| Scheme3 | Wikitext2, num=10 | [24, 11, 12] | 13.5906 | 52.0979 | 12.1825 | simu_num=8000 |
| | | [24, 11, 12, 27, 23, 8] | 17.7254 | 61.8678 | 15.1614 | epoch=300 |
| | | [24, 11, 12, 27, 23, 8, 21, 10, 14] | 25.3905 | 79.4398 | 20.4018 | mc=8000 |
| | | [24, 11, 12, 27, 23, 8, 21, 10, 14, 22, 25, 18] | 52.0979 | 139.4213 | 36.3702 | ks=(30,27,24,21,18) |
| Scheme4 | Bookcorpus, num=50 | [22, 21, 12] | 14.9263 | 52.9183 | 12.9682 | |
| | | [22, 21, 12, 24, 11, 23] | 19.7741 | 61.8678 | 15.8890 | |
| | | [22, 21, 12, 24, 11, 23, 14, 25, 10] | 25.7903 | 76.9957 | 20.7231 | |
| | | [22, 21, 12, 24, 11, 23, 14, 25, 10, 18, 17, 8] | 52.0979 | 139.4213 | 37.5247 | |
| Scheme5 | Bookcorpus, num=100 | [23, 11, 21] | 14.6949 | 53.7517 | 12.9682 | |
| | | [23, 11, 21, 12, 24, 25] | 18.2881 | 61.8678 | 15.8890 | |
| | | [23, 11, 21, 12, 24, 25, 14, 18, 10] | 24.9969 | 76.9957 | 20.7231 | |
| | | [23, 11, 21, 12, 24, 25, 14, 18, 10, 22, 8, 13] | 61.8678 | 268.7415 | 34.1666 | |

Table 20: Ablation study on calibration dataset for layer pruning in LLaMA-2-7B-hf.

# G   COMPUTATIONAL COST AND PRACTICAL OVERHEAD

We provide an overview of the computational cost of our pruning framework to give readers a sense of its efficiency in Tab. 23. Our method consists of two stages:

**Stage 1: Mask Evaluation.** In this stage, we evaluate the contribution of each layer by generating and scoring a set of pruning masks. For the LLaMA-2-7B-hf model, evaluating 8,000 randomly generated masks requires approximately 15 minutes on a single NVIDIA V100 GPU with 32GB of memory. This step establishes the performance landscape needed to estimate layer importance.

**Stage 2: Surrogate Model Training and Contribution Estimation.** Once the evaluation data is collected, we train a lightweight surrogate model to predict the performance of unseen masks. Training the surrogate model is extremely fast, taking less than one minute. Using the surrogate model to score 80,000 masks and estimate Shapley-based importance values for all layers requires around 15 minutes on the same V100 GPU.

Without the surrogate model (i.e., directly estimating Shapley values by evaluating each of the 80,000 masks across all 32 layers), the computation would require roughly $80,000 \times 32$ forward passes, taking about 320 hours on the same hardware. Furthermore, computing exact Shapley values without Monte Carlo approximation would require evaluating all possible combinations of layers, i.e., $2^{32}$ masks for a 32-layer model, which is computationally infeasible. This highlights the ne-

| Scheme | Simu_Num | Pruned Layer Index | PPL_WikiText2 | PPL_PTB | PPL_C4 | Other Setting |
|---|---|---|---|---|---|---|
| Scheme1 | 8000 | [21, 23, 11] | 14.6949 | 53.7517 | 12.9682 | |
| | | [21, 23, 11, 12, 18, 24] | 18.8686 | 61.8678 | 16.1392 | |
| | | [21, 23, 11, 12, 18, 24, 10, 27, 25] | 24.6093 | 76.9957 | 20.7231 | |
| | | [21, 23, 11, 12, 18, 24, 10, 27, 25, 14, 8, 9] | 38.1157 | 105.2407 | 28.7712 | |
| Scheme6 | 5000 | [11, 23, 21] | 14.6949 | 53.7517 | 12.9682 | |
| | | [11, 23, 21, 12, 25, 27] | 18.5761 | 61.8678 | 15.8890 | |
| | | [11, 23, 21, 12, 25, 27, 18, 10, 24] | 24.6093 | 76.9957 | 20.7231 | |
| | | [11, 23, 21, 12, 25, 27, 18, 10, 24, 13, 22, 14] | 54.5982 | 229.8668 | 35.8063 | |
| Scheme7 | 3000 | [21, 11, 24] | 14.4671 | 53.7517 | 12.7671 | epoch=200 |
| | | [21, 11, 24, 10, 8, 27] | 18.0046 | 63.8317 | 15.6426 | mc=80000 |
| | | [21, 11, 24, 10, 8, 27, 18, 12, 23] | 25.3905 | 76.9957 | 20.4018 | ks=(30,27,24,21,18) |
| | | [21, 11, 24, 10, 8, 27, 18, 12, 23, 25, 7, 14] | 45.9763 | 112.0281 | 31.5990 | |
| Scheme8 | 10000 | [11, 21, 23] | 14.6949 | 53.7517 | 12.9682 | |
| | | [11, 21, 23, 12, 27, 18] | 18.0046 | 61.8678 | 15.8890 | |
| | | [11, 21, 23, 12, 27, 18, 25, 10, 24] | 24.6093 | 76.9957 | 20.7231 | |
| | | [11, 21, 23, 12, 27, 18, 25, 10, 24, 9, 14, 17] | 43.8708 | 126.9445 | 32.0966 | |
| Scheme9 | 15000 | [11, 23, 21] | 14.6949 | 53.7517 | 12.9682 | |
| | | [11, 23, 21, 12, 10, 27] | 17.1801 | 61.8678 | 15.1614 | |
| | | [11, 23, 21, 12, 10, 27, 18, 24, 25] | 24.6093 | 76.9957 | 20.7231 | |
| | | [11, 23, 21, 12, 10, 27, 18, 24, 25, 9, 7, 14] | 43.8708 | 126.9445 | 32.0966 | |

Table 21: Ablation study on the number of Monte Carlo simulations (Simu_Num) for layer pruning in LLaMA-2-7B-hf.

| Scheme | Hamming Weight | Pruned Layer Index | PPL_WikiText2 | PPL_PTB | PPL_C4 | Other Setting |
|---|---|---|---|---|---|---|
| Scheme1 | ks=(30,27,24,21,18) | [21, 23, 11] | 14.6949 | 53.7517 | 12.9682 | |
| | | [21, 23, 11, 12, 18, 24] | 18.8686 | 61.8678 | 16.1392 | |
| | | [21, 23, 11, 12, 18, 24, 10, 27, 25] | 24.6093 | 76.9957 | 20.7231 | |
| | | [21, 23, 11, 12, 18, 24, 10, 27, 25, 14, 8, 9] | 38.1157 | 105.2407 | 28.7712 | |
| Scheme10 | ks=(30,27,24,21,10) | [11, 24, 12] | 13.6874 | 51.9857 | 12.2754 | |
| | | [11, 24, 12, 23, 10, 20] | 18.4794 | 60.9857 | 15.5452 | |
| | | [11, 24, 12, 23, 10, 20, 25, 21, 7] | 25.7113 | 77.5073 | 20.4523 | |
| | | [11, 24, 12, 23, 10, 20, 25, 21, 7, 14, 8, 27] | 48.6813 | 114.6653 | 33.0519 | |
| Scheme11 | ks=(30,27,14,12,10) | [11, 12, 23] | 13.8152 | 52.4852 | 12.3442 | simu_num=8000 |
| | | [11, 12, 23, 21, 24, 14] | 18.4125 | 61.4412 | 15.7563 | epoch=200 |
| | | [11, 12, 23, 21, 24, 14, 20, 18, 25] | 30.7862 | 84.7057 | 24.0289 | mc=80000 |
| | | [11, 12, 23, 21, 24, 14, 20, 18, 25, 10, 22, 7] | 55.5611 | 159.2974 | 37.7783 | |
| Scheme12 | ks=(30,16,14,12,10) | [6, 14, 20] | 15.455 | 55.6454 | 13.1504 | |
| | | [6, 14, 20, 8, 9, 21] | 21.5396 | 71.4415 | 16.8722 | |
| | | [6, 14, 20, 8, 9, 21, 10, 25, 26] | 32.214 | 88.0215 | 22.2093 | |
| | | [6, 14, 20, 8, 9, 21, 10, 25, 26, 15, 29, 23] | 74.3637 | 149.7612 | 48.9448 | |
| Scheme13 | Generate Mask Randomly | [19, 8, 16] | 16.3933 | 59.9643 | 14.0220 | |
| | | [19, 8, 16, 17, 28, 30] | 29.6845 | 95.8227 | 23.4824 | |
| | | [19, 8, 16, 17, 28, 30, 3, 12, 15] | 59.9643 | 190.5663 | 43.8708 | |
| | | [19, 8, 16, 17, 28, 30, 3, 12, 15, 23, 25, 7] | 115.5843 | 334.4542 | 76.9957 | |

Table 22: Ablation study on Hamming weight constraint for mask generation in LLaMA-2-7B-hf.

cessity of our two-stage approximation method, combining stratified Monte Carlo sampling with a lightweight surrogate model, to estimate layer importance efficiently.

| Scheme | Method | Computation | Approx. Time |
|---|---|---|---|
| Scheme 1 | Stage 1 + Stage 2 (Our method) | 8,000 + surrogate for 80,000 | 15 minutes + 15 minutes |
| Scheme 2 | Direct evaluation (Monte Carlo Shapley) | $80,000 \times 32$ forward passes | $\sim 5 \times 32$ hours |
| Scheme 3 | Exact Shapley computation | $2^{32}$ forward passes | Infeasible |

Table 23: Computational overhead for estimating layer importance on LLaMA-2-7B-hf using 32GB V100 GPU.

# H  INTEGRATION OF STRUCTURED PRUNING AND UNSTRUCTURED PRUNING

Here we demonstrate the effectiveness of combining structured and unstructured pruning methods, which leverages the strengths of both approaches. Specifically, unstructured pruning, as exemplified by SparseGPT, excels in maintaining high post-pruning accuracy but results in irregular sparse matrices, making it more suited for storage compression than inference acceleration. On the other hand,

structured pruning—such as depth pruning—provides a more regular sparsity pattern that can significantly accelerate inference. Our results in Tab.24 show that integrating these two methods strikes a balance between model performance and computational efficiency. Specifically, using LLaMA2-7B model as an example, we divide the pruning process into unstructure pruning and structure pruning. In the first stage, we use the SparseGPT method to prune weights. In the second stage, we compress the model obtained in the first stage by our proposed method. The experiment controlled the total pruning ratio at 37.5%, with the proportion of structure and non-structure being adjusted. The results show that increasing SparseGPT's pruning ratio while reducing our deep pruning ratio decreases perplexity (PPL) but reduces efficiency. Conversely, reducing PPL typically enhances efficiency.

| Integrated Method | | PPL | | | Efficiency | |
|---|---|---|---|---|---|---|
| Unstructured Ratio | Structured Ratio | WikiText2 | PTB | C4 | Latency(sec) | Throughout(tokens/s) |
| 0% | 100% | 38.1157 | 105.2407 | 28.7712 | 2.2141 | 57.8891 |
| 28% | 72% | 24.7917 | 76.1376 | 20.0237 | 2.4163 | 52.9739 |
| 50% | 50% | 18.2917 | 66.1394 | 16.1004 | 2.7269 | 46.9398 |
| 72% | 28% | 14.4598 | 53.9301 | 12.9936 | 3.044 | 42.1448 |
| 100% | 0% | 13.5921 | 50.4833 | 11.8936 | 3.3142 | 38.6315 |

Table 24: Integration of structured pruning and unstructured pruning. Key metrics such as Perplexity (PPL), Latency (sec), Throughput (tokens/s), and Number of Parameters (nparam) are analyzed across different pruning configurations.

It is worth noting that during our experiments, we also discovered that, while ensuring approximate performance and efficiency requirements, a combined approach can achieve a higher compression rate compared to using a single pruning method. For instance, when we prune the model using 100% structured pruning (without any unstructured pruning), the resulting model with approximately 5.5B parameters achieves perplexity (PPL) values of 18.8686, 61.8678, and 16.1392 on the WikiText2, PTB, and C4 datasets, respectively, with inference latency of 2.7554 seconds and throughput of 46.455 tokens/s. However, when we apply a combination of 26% unstructured pruning and 74% structured pruning, reducing the model to 4.6B parameters, we are able to maintain similar PPL values and inference efficiency.

| Params | Integrated Prune | | PPL | | | Efficiency | |
|---|---|---|---|---|---|---|---|
| | Unstructured Ratio | Structured Ratio | PPL_WikiText2 | PPL_PTB | PPL_C4 | Latency(sec) | Throughout(tokens/s) |
| 5.5B | 0% | 100% | 18.8686 | 61.8678 | 16.1392 | 2.7554 | 46.455 |
| 5.0B | 14% | 86% | 18.3585 | 64.8451 | 15.6334 | 2.7678 | 46.249 |
| 4.6B | 26% | 74% | 18.2917 | 66.1394 | 16.1004 | 2.7269 | 46.9398 |
| 4.1B | 44% | 56% | 19.6860 | 63.9139 | 16.2838 | 2.7237 | 46.9953 |

Table 25: Integration of structured pruning and unstructured pruning. Key metrics such as Perplexity (PPL), Latency (sec), Throughput (tokens/s), and Number of Parameters (nparam) are analyzed across different pruning configurations.

We have added a comparison with the SLEB method in Tab.26. Specifically, using the LLaMA2-7B model as an example, we first sparsified the model using SparseGPT and then performed further depth-wise pruning on the sparsified model using both our method and SLEB. To ensure the robustness of the experimental results, we used four models with sparse ratios of 0.1, 0.18, 0.27, and 0.36, and performed depth pruning with 6, 9, and 12 layers pruned. For a fair comparison, the experiments were conducted under the same settings. The experimental results in the table below show that our method outperforms SLEB when integrated with the unstructured pruning approach.

| Unstructured Method | Structured Method | PPL_WikiText2 | | PPL_C4 | |
|---|---|---|---|---|---|
| Sparse Rate | Remove layer counts | Ours | SLEB | Ours | SLEB |
| | 6 | **18.3585** | 19.4312 | **15.6334** | 16.3469 |
| 0.1 | 9 | **24.7917** | 27.3805 | **20.0237** | 21.5788 |
| | 12 | **52.8705** | 58.9879 | **32.1339** | 44.3139 |
| | 6 | **18.2917** | 19.7478 | **16.1004** | 16.5431 |
| 0.18 | 9 | **27.1824** | 27.8619 | **21.8909** | 21.9230 |
| | 12 | **53.7073** | 59.6368 | **37.2207** | 44.6520 |
| | 6 | **19.6860** | 23.8295 | **16.2838** | 18.8977 |
| 0.27 | 9 | **26.6423** | 59.7167 | **21.5267** | 43.5821 |
| | 12 | **41.7826** | 189.4915 | **30.9364** | 135.0492 |
| | 6 | **20.6773** | 22.9696 | **17.472** | 18.6676 |
| 0.36 | 9 | **30.6545** | 33.6778 | **23.3234** | 25.4039 |
| | 12 | **59.6735** | 75.7457 | **37.5587** | 55.5516 |

Table 26: Integration of structured pruning and unstructured pruning. Key metrics such as Perplexity (PPL), Latency (sec), Throughput (tokens/s), and Number of Parameters (nparam) are analyzed across different pruning configurations.

