# OpenReview forum: "Pruning as a Cooperative Game: Surrogate-Assisted Layer Contribution Estimation for Large Language Models"
_ICLR.cc/2026/Conference — ICLR 2026 Poster_

### Official Review · Reviewer_jT3z · 2025-10-28

**Soundness:** 3
**Presentation:** 3
**Contribution:** 2
**Rating:** 6
**Confidence:** 4

**Summary:**

The paper points out that existing layer pruning methods mostly rely on static heuristic rules, overlooking the dynamic interdependencies between layers, which leads to suboptimal results. To address this issue, the authors propose a game-theoretic approach that formulates the layer pruning problem as a cooperative game. In this game, each Transformer layer is treated as a “player,” and the overall model performance (measured by perplexity, PPL) represents the collective “utility” produced through cooperation among all players. Since computing the exact contribution of each player (i.e., the Shapley value) is computationally intractable, the authors further design an efficient two-stage approximation framework to estimate these contributions and prune layers with lower importance. Experimental results show that this method consistently and significantly outperforms existing depth-wise and width-wise pruning baselines on both language modeling (PPL) and zero-shot reasoning tasks. Moreover, it generalizes well to non-Transformer architectures and demonstrates strong compatibility with quantization techniques such as GPTQ.

**Strengths:**

1. The motivation is clear, and the results in Figure 1 clearly demonstrate the interdependence among layers.
2. The paper proposes an efficient two-stage approximation framework that significantly reduces the computational complexity of solving the cooperative game.
3. The experimental evaluation is comprehensive, covering both Transformer and non-Transformer model architectures.

**Weaknesses:**

1. The paper’s core innovation is insufficient. The idea of viewing pruning from a cooperative game theory perspective has already been explored in prior work, such as “Using Cooperative Game Theory to Prune Neural Networks.” In addition, “Draft & Verify: Lossless Large Language Model Acceleration via Self-Speculative Decoding” also formulates the layer pruning problem as an optimization task through binary pruning masks.
2. The experimental validation is not sufficiently comprehensive. The benchmarks used are limited to perplexity (PPL) and multiple-choice tasks, without evaluation on generation benchmarks such as GSM8K. In the generalization experiments on non-Transformer architectures, only PPL was reported, lacking broader task evaluation.

**Questions:**

1. Why did the authors choose to evaluate on the ANLI benchmark?
2. Why was MMLU not included in the evaluation? According to the results reported in the ShortGPT paper, its pruning method performs well on MMLU. Given that MMLU is a standard benchmark for evaluating reasoning and knowledge retention in large language models, the authors should at least include results on this task to make the evaluation more complete.
3. Regarding the implementation of iterative pruning, the paper mentions “iteratively removing the least contributive layers.” Could the authors clarify how this iterative process is carried out?
(A) Are all layers’ Shapley values computed once, and then layers are removed in batches (e.g., first pruning the three least contributive layers, then the next three)?
(B) Or after each batch of layers is removed (e.g., three layers), are both Stage 1 and Stage 2 rerun to recompute the Shapley values for the remaining layers?
If the process follows (A), it seems inconsistent with the paper’s main motivation that pruning changes the relative importance of other layers. If it follows (B), the overall computational cost would increase significantly.

---

> ### Author Response · Authors · 2025-11-23
> **Response to Reviewer jT3z**
>
> We sincerely thank you for the thorough review of our paper and for your further suggestion. We provide our feedbacks as follows.
>
> ---
> ## Weakness 1
> > The idea of viewing pruning from a cooperative game theory perspective has already been explored in prior work, such as “Using Cooperative Game Theory to Prune Neural Networks.” “Draft \& Verify: Lossless Large Language Model Acceleration via Self-Speculative Decoding” also formulates the layer pruning problem as an optimization task through binary pruning masks.
>
> **Answer to W1:** Thank you for your valuable feedback and for highlighting related work. Our work shares similarities with "Using Cooperative Game Theory to Prune Neural Networks" in using a game-theoretic perspective, but differs in problem definition, method granularity, and application. Speicifically, the method proposed in that paper focuses on evaluating individual neuron contributions, which is impractical for large models like LLMs due to exponential growth in neurons. In contrast, our method performs game-theoretic layer-level analysis in LLMs and using a pre-trained surrogate model to predict performance changes after pruning, enabling efficient sampling and avoiding high evaluation costs. Regarding "Draft \& Verify: Lossless Large Language Model Acceleration via Self-Speculative Decoding," while both methods aim to accelerate LLM inference, our approach permanently prunes layers to optimize the model architecture offline, without requiring additional validation steps during inference, as D\&V does. Additionally, D\&V uses Bayesian optimization for skip-layer combinations, which is a black-box optimization process, while our method uses Shapley values to assess layer importance, providing a fair and accurate measurement of each layer's contribution to the model’s predictive task.
>
> ---
> ## Weakness 2
> > The experimental validation is not sufficiently comprehensive.
>
> **Answer to W1:** Thank you for your valuable feedback. We appreciate your suggestion for expanding the experimental validation to include additional benchmarks. We chose perplexity (PPL) and multiple-choice tasks as primary benchmarks to evaluate the pruned models, as they are commonly used to assess the performance of large language models, especially in the context of model compression. However, we agree that including additional generation benchmarks would provide a broader view of the model's capabilities. We have added the MMLU benchmark in our evaluation. The results have been updated in the revised PDF, and you can refer to Table 3 for the detailed evaluation.
>
> ---
> ## Question 1
> > Why did the authors choose to evaluate on the ANLI benchmark?
>
> **Answer to Q1:** We chose the ANLI benchmark to evaluate the pruned model’s generalization performance on adversarial data. ANLI, with adversarially generated natural language inference tasks, tests the model’s ability to understand subtle and complex language relations. In real-world applications, models often encounter ambiguous or contradictory inputs, making this evaluation necessary for assessing the robustness of pruned models.
>
> ---
> ## Question 2
> > Why was MMLU not included in the evaluation? According to the results reported in the ShortGPT paper, its pruning method performs well on MMLU. Given that MMLU is a standard benchmark for evaluating reasoning and knowledge retention in large language models, the authors should at least include results on this task to make the evaluation more complete.
>
> **Answer to Q2:** Thank you for your valuable question. As mentioned in Answer to W2, we have added the MMLU benchmark in our evaluation. The results have been updated in the revised PDF, and you can refer to Table 3 for the detailed evaluation.
>
> ---
> ## Question 3
> > Regarding the implementation of iterative pruning, the paper mentions “iteratively removing the least contributive layers.” Could the authors clarify how this iterative process is carried out? (A) Are all layers’ Shapley values computed once, and then layers are removed in batches? (B) Or after each batch of layers is removed, are both Stage 1 and Stage 2 rerun to recompute the Shapley values for the remaining layers?
>
> **Answer to Q3:** We apologize for the confusion caused by the inappropriate use of the term "iteratively" in describing the pruning process in the paper. To clarify, our method involves pruning the layers with the smallest contributions in sequence, after calculating the importance of all layers, until the desired pruning ratio is reached. This approach is based on the Shapley values, which we compute using the surrogate model, and thus account for the interdependencies between layers. Therefore, the pruning process can directly rely on these values. To ensure clarity, we have updated the description in the revised PDF to reflect this correction.

---

### Official Review · Reviewer_CfBF · 2025-10-28

**Soundness:** 3
**Presentation:** 3
**Contribution:** 2
**Rating:** 6
**Confidence:** 4

**Summary:**

This paper introduces a game-theoretic framework for pruning LLMs, aiming to reduce computational cost while preserving model performance. Instead of treating layers independently, the authors model pruning as a cooperative game, where each transformer layer is a “player” and the model’s performance serves as the utility function.

**Strengths:**

1. This paper reformulates LLM pruning as a cooperative game, capturing inter-layer dependencies ignored by static heuristics.
2. This paper proposes a scalable Shapley-based pruning framework using stratified sampling and a surrogate model for efficient layer contribution estimation.
3. This paper demonstrates consistent improvements over depth- and width-wise pruning baselines on multiple benchmarks, including WikiText2, PTB, C4, and zero-shot reasoning tasks.

**Weaknesses:**

1. While the paper proposes a surrogate-assisted approach to estimate Shapley values efficiently, it lacks a clear theoretical analysis quantifying how well the surrogate approximates true layer contributions.
2. The method relies on a small calibration set, which may not adequately represent diverse data distributions or downstream task requirements. The resulting Shapley estimates might therefore be dataset-dependent and unstable across domains.
3. The study focuses solely on one-shot pruning without retraining, which may restrict achievable performance. Many recent works (e.g., ShortGPT) benefit from lightweight fine-tuning. It will be helpful to investigate how minor retraining after pruning interacts with the cooperative-game framework.

**Questions:**

1. The paper uses 10 BookCorpus samples for calibration. How was this number chosen, and how does increasing or diversifying the calibration set affect Shapley estimation quality?
2. Have you explored whether minimal fine-tuning after pruning further improves performance?
3. Have you profiled the wall-clock speedups and memory reductions on real hardware (e.g., A100, H100, or consumer GPUs)? How does the pruning affect model latency when combined with quantization in real-time inference settings?

---

> ### Author Response · Authors · 2025-11-23
> **Response to Reviewer CfBF (Part 1)**
>
> We thank reviewer for the constructive comments. We provide our feedbacks as follows.
>
> ---
> ## Weakness 1
> > It lacks a clear analysis quantifying how well the surrogate approximates true layer contributions.
>
> **Answer to W1:** Thank you for your valuable feedback. We evaluated the surrogate model's performance using the LLaMA-2-7B-hf model with the training mask ks=(30,27,24,21,18), corresponding to our predefined Hamming weight settings, and analyzed the R² metric. The results show that the surrogate model performs consistently across different random seeds and test sample sizes. For various mask configurations, we systematically varied the Hamming weight of the test mask. The results demonstrate that the surrogate model maintains high predictive accuracy when the test mask closely matches the training mask. However, when the test mask differs significantly from the training mask, the model’s performance declines, as expected. In our formal experiments, we will adjust the mask according to the desired pruning ratio, ensuring the test mask remains similar to the training mask. We have included the detailed experimental results in Appendix B.1 (Table 8) of the revised manuscript.
>
> |            Case            | Test Samples  | Seed  |         Test Mask          | R$^2$  |
> |:--------------------------|:-------------:|:-----:|:--------------------------:|:------:|
> | Same Seed, Same Mask  |      200      |   42  | ks = (30, 27, 24, 21, 18)  |  0.9360 |
> |                            |      500      |   42  | ks = (30, 27, 24, 21, 18)  | 0.9492 |
> |                            |      1000     |   42  | ks = (30, 27, 24, 21, 18)  | 0.9464 |
> | Different Seeds, Same Mask |      500      |  500  | ks = (30, 27, 24, 21, 18)  | 0.9359 |
> |                            |      500      |  1234 | ks = (30, 27, 24, 21, 18)  | 0.9402 |
> |                            |      500      | 99999 | ks = (30, 27, 24, 21, 18)  |  0.9400  |
> | Same Seed, Different Masks |      500      |   42  |  ks = (16, 27, 24, 21, 18) |  0.9230 |
> |                            |      500      |   42  |  ks = (16, 14, 24, 21, 18) | 0.8658 |
> |                            |      500      |   42  |  ks = (16, 14, 12, 21, 18) | 0.2466 |
>
> ---
> ## Weakness 2
> > The method relies on a small calibration set, which may not adequately represent diverse data distributions or downstream task requirements. The resulting Shapley estimates might therefore be dataset-dependent and unstable across domains.
>
> **Answer to W2:** Thank you for your insightful feedback. The choice of using 10 BookCorpus samples for calibration was based on the settings used in the Shortened-LLAMA paper, ensuring consistency and fairness in experimental comparisons. To make sure all methods requiring calibration datasets were evaluated under similar conditions, we standardized this setup across all experiments. Additionally, we conducted ablation experiments on the calibration dataset size and its impact on model performance. The results of these experiments are detailed in Appendix F.1, with the corresponding results shown in Table 20. These experiments demonstrate that while the calibration set is relatively small, the surrogate model's performance remains stable across various dataset configurations, supporting the generalizability of our approach.
>
> ---
> ## Weakness 3
> > The study focuses solely on one-shot pruning without retraining, which may restrict achievable performance. Many recent works (e.g., ShortGPT) benefit from lightweight fine-tuning. It will be helpful to investigate how minor retraining after pruning interacts with the cooperative-game framework.
>
> **Answer to W3:** Thank you for your valuable question. Due to the page limits, we have placed the experiments on pruning followed by fine-tuning in Appendix. Specifically, in Appendix E, we used LoRA to fine-tune the pruned model, and Figure 7 shows the performance improvement of the pruned model after fine-tuning.
>
> ---
> ## Question 1
> > The paper uses 10 BookCorpus samples for calibration. How was this number chosen, and how does increasing or diversifying the calibration set affect Shapley estimation quality?
>
> **Answer to Q1:** Thank you for your insightful question. As mentioned in Answer to W2, the paper's choice of 10 BookCorpus was based on the settings of the shortened-llama article, and to ensure the fairness of experimental comparisons, we standardised the above settings for all methods that require calibration datasets. We conducted ablation experiments on the calibration datasets in Appendix F.1 of the original text, and the experimental results are shown in Table 20.

---

> ### Author Response · Authors · 2025-11-23
> **Response to Reviewer CfBF (Part 2)**
>
> ## Question 2
> > Have you explored whether minimal fine-tuning after pruning further improves performance?
>
> **Answer to Q2:** As mentioned in Answer to W3, we have placed the experiments on pruning followed by fine-tuning and the discussion on computational costs in Appendix. Specifically, in Appendix E, we used LoRA to fine-tune the pruned model, and Figure 7 shows the performance improvement of the pruned model after fine-tuning.
>
> ---
> ## Question 3
> > Have you profiled the wall-clock speedups and memory reductions on real hardware (e.g., A100, H100, or consumer GPUs)? How does the pruning affect model latency when combined with quantization in real-time inference settings?
>
> **Answer to Q3:** Thank you for your thoughtful question. We provide a detailed discussion of the runtime efficiency of our method in Appendix G, where we evaluate its performance on a 32GB V100 GPU. Regarding the impact of pruning combined with quantization on real-time inference latency, we have addressed this concern in Section 4.6 of the paper. Specifically, we evaluate how pruning affects model latency when combined with quantization, and the results are shown in Figure 5. These results demonstrate the improvements in inference speed due to pruning and the effect of quantization on reducing model size in real-time inference scenarios.

---

### Official Review · Reviewer_FgAt · 2025-10-31

**Soundness:** 3
**Presentation:** 2
**Contribution:** 2
**Rating:** 4
**Confidence:** 4

**Summary:**

This paper proposes a game-theoretic approach to layer-wise pruning in large language models, casting the problem as a cooperative game where each layer is a player and model performance serves as utility. Since computing exact Shapley values to measure each layer’s marginal contribution is infeasible at scale, the authors design a two-stage approximation using stratified Monte Carlo mask sampling and a lightweight surrogate network to efficiently estimate layer importance. The framework reportedly captures inter-layer dependencies better than prior methods and consistently surpasses strong depth-wise and width-wise pruning baselines across multiple downstream and generative benchmarks for Transformer and non-Transformer models.

**Strengths:**

- **Principled Formulation & Theoretical Motivation**: The paper introduces a compelling game-theoretic framing for layer pruning, challenging the prevalent assumption of independent layer importance and instead recognizing context-dependent inter-layer dynamics. This principled approach addresses a core limitation of widely-used heuristics.
- **Efficient Approximation of Shapley Values**: By incorporating a lightweight surrogate network trained on stratified Monte Carlo mask samples, the method approximates Shapley values efficiently, enabling practical application to large-scale LLMs—this is articulated with clear algorithmic details and demonstrated scalability (see Algorithm 1 & Figure 2).
- **Conceptual Generality**: The approach is validated across Transformer and non-Transformer architectures, and shown to integrate compatibly with quantization and LoRA fine-tuning (Figure 7), suggesting broad applicability for LLM deployment.

**Weaknesses:**

- **Surrogate Network Limitations and Validation**: While Figure 6 and Table 7 specify the surrogate’s structure, the paper lacks a rigorous quantitative evaluation of its prediction fidelity, especially for masks far from the training distribution. There is little discussion of failure modes, e.g., overfitting to calibration samples, brittleness under extreme masking, or calibration data misspecification (see Appendix F.1), limiting confidence for highly compressed regimes or out-of-domain settings. For instance, the impact of surrogate error on Shapley ranking stability is not systematically assessed.
- **Potential Optimization and Mask Generation Shortcomings**: The stratified Monte Carlo mask sampling strategy (Section 3.3, Table 21) is justified primarily by ablation, with theoretical explanations on sampling sufficiency or representativeness lacking. While empirical results (Table 21) suggest an advantage over random sampling, the method’s robustness to choice of Hamming weights, number of samples, and potential for bias due to nonuniform coverage of important layer subsets remains underexplored. The mask set’s coverage and the potential for missed critical coalitions, particularly as $L$ increases, are not discussed in depth.
- **Missing Related Work and Baselines**: The paper mentions SparseGPT but the citation to SparseGPT is pointing to the wrong paper (from the same research group).  Adding SparseGPT to the results would also make the experiments more complete. Furthermore, there has been research using Shapely values [1,2] and Influence Functions [3] for LLM pruning and LLM layer importance estimation that the paper fails to acknowledge.
- **Ambiguity in Theoretical Guarantees**: While the game-theoretic formulation is elegant, there is no quantification of the approximation gap between true Shapley values and those estimated by the surrogate. No guarantees or bounds are provided regarding the surrogate’s reliability or the stability of resulting pruning strategies as masking regimes change. This undermines full confidence in the method’s reliability for critical compression tasks.
- **Additional Minor Concerns**: Certain tables (such as Table 2) require careful reading to parse due to excessive fragmentation of sub-columns; legend clarity could be improved (see also Figure 3, Figure 4); some experimental settings (e.g., LoRA fine-tuning, ablation tasks) are relegated to the appendix without high-level results in the main paper.

**Questions:**

- How robust is the surrogate model to mismatches between the calibration data distribution and the actual deployment or test distribution? Have the authors quantified the surrogate’s error rate, particularly for masks not seen during training?

- Can the authors provide theoretical or empirical insights into the approximation gap between surrogate-predicted and true Shapley values, especially regarding stability of the ranking under different sampling or masking regimes?

- Can authors provide results for SparseGPT as well and acknowledge or compare against the newer pruning methods that were left out.

- Could the authors provide more interpretability on which types of layers (e.g., attention, feed-forward, early vs. late) are most frequently pruned, and relate this to observed task degradations

---

> ### Author Response · Authors · 2025-11-23
> **Response to Reviewer FgAt (Part 1)**
>
> We would like to thank you for your thoughtful comments. We hope that our response can address your questions. Please let us know if there are any further questions.
>
> ---
> ## Weakness 1
> > Lacks a rigorous quantitave evaluation of its prediction fidelity, especially for masks far from the training distribution.
>
> **Answer to W1:** Thank you for your valuable feedback. We evaluated the surrogate model's performance using the LLaMA-2-7B-hf model with the training mask ks=(30,27,24,21,18), corresponding to our predefined Hamming weight settings, and analyzed the R² metric. The results show that the surrogate model performs consistently across different random seeds and test sample sizes. For various mask configurations, we systematically varied the Hamming weight of the test mask. The results demonstrate that the surrogate model maintains high predictive accuracy when the test mask closely matches the training mask. However, when the test mask differs significantly from the training mask, the model’s performance declines, as expected. In our formal experiments, we will adjust the mask according to the desired pruning ratio, ensuring the test mask remains similar to the training mask. We have included the detailed experimental results in Appendix B.1 (Table 8) of the revised manuscript.
>
> |            Case            | Test Samples  | Seed  |         Test Mask          | R$^2$  |
> |:--------------------------|:-------------:|:-----:|:--------------------------:|:------:|
> | Same Seed, Same Mask  |      200      |   42  | ks = (30, 27, 24, 21, 18)  |  0.9360 |
> |                            |      500      |   42  | ks = (30, 27, 24, 21, 18)  | 0.9492 |
> |                            |      1000     |   42  | ks = (30, 27, 24, 21, 18)  | 0.9464 |
> | Different Seeds, Same Mask |      500      |  500  | ks = (30, 27, 24, 21, 18)  | 0.9359 |
> |                            |      500      |  1234 | ks = (30, 27, 24, 21, 18)  | 0.9402 |
> |                            |      500      | 99999 | ks = (30, 27, 24, 21, 18)  |  0.9400  |
> | Same Seed, Different Masks |      500      |   42  |  ks = (16, 27, 24, 21, 18) |  0.9230 |
> |                            |      500      |   42  |  ks = (16, 14, 24, 21, 18) | 0.8658 |
> |                            |      500      |   42  |  ks = (16, 14, 12, 21, 18) | 0.2466 |

---

> ### Author Response · Authors · 2025-11-23
> **Response to Reviewer FgAt (Part 2)**
>
> ## Weaknesss 2
>
> > Potential Optimization and Mask Generation Shortcomings.
>
> **Answer to W2:** Thank you for your valuable suggestion. To further explore the impact of the Hamming weight constraint, we conducted additional experiments using the LLaMA-2-7B-hf model with various Hamming weight settings. Specifically, Scheme 1 corresponds to our main experimental setting, where we use predefined Hamming weights $ks=(30, 27, 24, 21, 18)$ for stratified sampling. We also tested other configurations (Scheme 2, Scheme 3, and Scheme 4) where the Hamming weight values were adjusted incrementally. Scheme 5 represents a fully random mask sampling method. The results show that the choice of Hamming weight range has a significant impact on pruning performance. As we intervene more with the Hamming weights (from Scheme 2 to Scheme 4), the pruning performance progressively worsens, and this effect becomes more pronounced as the pruning depth increases. However, all constrained sampling schemes (Schemes 1, 2, 3, and 4) outperform the random sampling (Scheme 5). The best performance achieved by Scheme 1 further suggests that the Hamming weight range should be adjusted based on the desired pruning ratio. Therefore, selecting a balanced Hamming weight range that aligns with the pruning requirements is crucial for maintaining performance and ensuring diverse and effective pruning patterns. We have added the experimental results in Appendix F.3 of the revised manuscript, with corresponding analysis provided in Table 22.
>
> | Scheme  | Hamming Weight         | Pruned Layer Index                              | PPL_WikiText2 |  PPL_PTB |  PPL_C4 |
> |---------|------------------------|-------------------------------------------------|:-------------:|:--------:|:-------:|
> | Scheme1 | ks=(30,27,24,21,18)    | [21, 23, 11]                                    |    14.6949    |  53.7517 | 12.9682 |
> |         |                        | [21, 23, 11, 12, 18, 24]                        |    18.8686    |  61.8678 | 16.1392 |
> |         |                        | [21, 23, 11, 12, 18, 24, 10, 27, 25]            |    24.6093    |  76.9957 | 20.7231 |
> |         |                        | [21, 23, 11, 12, 18, 24, 10, 27, 25, 14, 8, 9]  |    38.1157    | 105.2407 | 28.7712 |
> | Scheme2 | ks=(30,27,24,21,10)    | [11, 24, 12]                                    |    13.6874    |  51.9857 | 12.2754 |
> |         |                        | [11, 24, 12, 23, 10, 20]                        |    18.4794    |  60.9857 | 15.5452 |
> |         |                        | [11, 24, 12, 23, 10, 20, 25, 21, 7]             |    25.7113    |  77.5073 | 20.4523 |
> |         |                        | [11, 24, 12, 23, 10, 20, 25, 21, 7, 14, 8, 27]  |    48.6813    | 114.6653 | 33.0519 |
> | Scheme3 | ks=(30,27,14,12,10)    | [11, 12, 23]                                    |    13.8152    |  52.4852 | 12.3442 |
> |         |                        | [11, 12, 23, 21, 24, 14]                        |    18.4125    |  61.4412 | 15.7563 |
> |         |                        | [11, 12, 23, 21, 24, 14, 20, 18, 25]            |    30.7862    |  84.7057 | 24.0289 |
> |         |                        | [11, 12, 23, 21, 24, 14, 20, 18, 25, 10, 22, 7] |    55.5611    | 159.2974 | 37.7783 |
> | Scheme4 | ks=(30,16,14,12,10)    | [6, 14, 20]                                     |     15.455    |  55.6454 | 13.1504 |
> |         |                        | [6, 14, 20, 8, 9, 21]                           |    21.5396    |  71.4415 | 16.8722 |
> |         |                        | [6, 14, 20, 8, 9, 21, 10, 25, 26]               |     32.214    |  88.0215 | 22.2093 |
> |         |                        | [6, 14, 20, 8, 9, 21, 10, 25, 26, 15, 29, 23]   |    74.3637    | 149.7612 | 48.9448 |
> | Scheme5 | Generate Mask Randomly | [19, 8, 16]                                     |    16.3933    |  59.9643 | 14.0220 |
> |         |                        | [19, 8, 16, 17, 28, 30]                         |    29.6845    |  95.8227 | 23.4824 |
> |         |                        | [19, 8, 16, 17, 28, 30, 3, 12, 15]              |    59.9643    | 190.5663 | 43.8708 |
> |         |                        | [19, 8, 16, 17, 28, 30, 3, 12, 15, 23, 25, 7]   |    115.5843   | 334.4542 | 76.9957 |
>
> ---
> ## Weakness 3
> > Missing Related Work and Baselines.
>
> **Answer to W3:** Thank you for your valuable suggestions. Based on your feedback, we have reviewed the citation of SparseGPT in the paper. Additionally, we have expanded the discussion of relevant literature on research using game theory to provide a more comprehensive overview. If you could kindly provide specific papers that you think we should consider, we would be happy to review them and incorporate relevant insights into our work.

---

> > ### Comment · Reviewer_FgAt · 2025-11-28
> > **Reply to rebuttal**
> >
> > My apologies for not properly attaching the references that I had cited. The papers are below:
> >
> > [1] Sun, Chuan, et al. "Efficient shapley value-based non-uniform pruning of large language models." arXiv preprint arXiv:2505.01731 (2025).
> > [2] Zhang, Yang, Yanfei Dong, and Kenji Kawaguchi. "Investigating layer importance in large language models." arXiv preprint arXiv:2409.14381 (2024).
> > [3] Askari, Hadi, et al. "LayerIF: Estimating Layer Quality for Large Language Models using Influence Functions." arXiv preprint arXiv:2505.23811 (2025).

---

> > > ### Comment · Reviewer_FgAt · 2025-11-28
> > > **Reply to rebuttal**
> > >
> > > Also, with regards to the SparseGPT experiments I would like to see:
> > >
> > > 1) The zero-shot accuracy numbers when just using SparseGPT and just using your method.
> > > 2) A similar structured pruning ratio to unstructured pruning ratio comparison with the other baselines and SparseGPT similar to the analysis that you performed with your method and SparseGPT. Only SLEB is also fine if you are short on time.

---

> > > > ### Author Response · Authors · 2025-12-03
> > > > **Further Response to Reviewer FgAt**
> > > >
> > > > Thank you for your insightful suggestion. In Table 24 of the appendix, the unstructured pruning ratios of 100\% and 0\% correspond to the results of compressing the model by 37.5\% using only SparseGPT and using only our method, respectively. As shown in the results, unstructured pruning achieves relatively high performance; however, its throughout is low and it cannot effectively realize acceleration. This observation is consistent with findings in existing pruning literature, where unstructured pruning generally incurs lower performance degradation but does so at the cost of reduced efficiency. According to your suggestion, we have added a comparison with the SLEB method. Specifically, using the LLaMA2-7B model, we first sparsified the model with SparseGPT, then applied further depth-wise pruning on the sparsified model with both our method and SLEB. To ensure the robustness of the experimental results, we tested four models with sparse ratios of 0.1, 0.18, 0.27, and 0.36, and performed depth pruning with 6, 9, and 12 layers pruned. All experiments were conducted under the same conditions for a fair comparison. The experimental results below show that our method outperforms SLEB when integrated with the unstructured pruning approach. We have added the experimental analysis in Appendix H of the revised manuscript, with corresponding results provided in Table 26.
> > > >
> > > > | **Unstructured Method** | **Structured Method** | **PPL_WikiText2** |          |  **PPL_C4** |          |
> > > > |:-----------------------:|:---------------------:|:-----------------:|:--------:|:-----------:|:--------:|
> > > > |       Sparse Rate       |  Remove layer counts  |        Ours       |   SLEB   |     Ours    |   SLEB   |
> > > > |           0.1           |           6           |    **18.3585**    |  19.4312 | **15.6334** |  16.3469 |
> > > > |                         |           9           |    **24.7917**    |  27.3805 | **20.0237** |  21.5788 |
> > > > |                         |           12          |    **52.8705**    |  58.9879 | **32.1339** |  44.3139 |
> > > > |           0.18          |           6           |    **18.2917**    |  19.7478 | **16.1004** |  16.5431 |
> > > > |                         |           9           |    **27.1824**    |  27.8619 | **21.8909** |  21.9230 |
> > > > |                         |           12          |    **53.7073**    |  59.6368 | **37.2207** |  44.6520 |
> > > > |           0.27          |           6           |    **19.6860**    |  23.8295 | **16.2838** |  18.8977 |
> > > > |                         |           9           |    **26.6423**    |  59.7167 | **21.5267** |  43.5821 |
> > > > |                         |           12          |    **41.7826**    | 189.4915 | **30.9364** | 135.0492 |
> > > > |           0.36          |           6           |    **20.6773**    |  22.9696 |  **17.472** |  18.6676 |
> > > > |                         |           9           |    **30.6545**    |  33.6778 | **23.3234** |  25.4039 |
> > > > |                         |           12          |    **59.6735**    |  75.7457 | **37.5587** |  55.5516 |

---

> > > ### Author Response · Authors · 2025-12-03
> > > **Further Response to Reviewer FgAt**
> > >
> > > Thank you for your valuable feedback. Based on the references you provided, we have added a discussion of these papers in the Related Work section. The revised version of the paper has been updated accordingly in the new PDF.

---

> ### Author Response · Authors · 2025-11-23
> **Response to Reviewer FgAt (Part 3)**
>
> ## Weakness 4
> > There is no quantification of the approximation gap between true Shapley values and those estimated by the surrogate. No guarantees or bounds are provided regarding the surrogate’s reliability or the stability of resulting pruning strategies as masking regimes change. This undermines full confidence in the method’s reliability for critical compression tasks.
>
> **Answer to W4:** Thank you for your insightful feedback. As mentioned in Answer to W1, we have added the corresponding experimental results and updated the PDF. The experiments demonstrate the surrogate model's prediction accuracy and show that it performs well across different seed values, test sample sizes, and mask distributions.
>
> ---
> ## Weakness 5
> > Additional Minor Concerns: Certain tables (such as Table 2) require careful reading to parse due to excessive fragmentation of sub-columns; legend clarity could be improved (see also Figure 3, Figure 4); some experimental settings (e.g., LoRA fine-tuning, ablation tasks) are relegated to the appendix without high-level results in the main paper.
>
> **Answer to W5:** Thank you for your valuable comments. Due to page limitations, in the previous submission we placed some experimental details in the appendix, and the layout of the tables may have been somewhat scattered. We understand the importance of these details for the readability of the paper and plan to make further adjustments and improvements in the final version. Thank you for pointing out these issues, which is very helpful for improving the manuscript.
>
> ---
> ## Question 1
> > How robust is the surrogate model to mismatches between the calibration data distribution and the actual deployment or test distribution? Have the authors quantified the surrogate’s error rate, particularly for masks not seen during training?
>
> **Answer to Q1:** Thank you for your insightful feedback. As mentioned in Answer to W1, we have added the corresponding experimental results and updated the PDF. The experiments demonstrate the surrogate model's prediction accuracy and show that it performs well across different seed values, test sample sizes, and mask distributions.
>
> ---
> ## Question 2
> > Can the authors provide theoretical or empirical insights into the approximation gap between surrogate-predicted and true Shapley values, especially regarding stability of the ranking under different sampling or masking regimes?
>
> **Answer to Q2:** Thank you for your insightful question. As mentioned in Answer to W1, we have added the corresponding experimental results in the updated PDF. These experiments assess the surrogate model’s prediction accuracy and show that it performs well across different seed values, test sample sizes, and mask distributions. We acknowledge that under different sampling and masking regimes, the ranking of layers can change somewhat. However, the final pruning results show minimal differences in terms of performance. This is expected, as our method takes into account the interdependencies between different layers, which helps mitigate the impact of variations in ranking. These results highlight the robustness of our approach and its ability to maintain high performance despite fluctuations in the layer rankings.

---

> ### Author Response · Authors · 2025-11-23
> **Response to Reviewer FgAt (Part 4)**
>
> ## Question 3
> > Can authors provide results for SparseGPT as well and acknowledge or compare against the newer pruning methods that were left out.
>
> **Answer to Q3:** Thank you for your valuable suggestions. SparseGPT is a key work on unstructured pruning, where weight importance is determined using the OBS error formula to decide which weights to prune. However, unstructured pruning often leads to irregularly structured sparse matrices, which are better suited for storage compression rather than inference acceleration. Our paper demonstrates that depth pruning, a form of structured pruning, can accelerate inference while preserving model performance. We believe that structured and unstructured pruning are complementary, and when combined, they can effectively enhance performance. In the updated PDF, we have added an experiment that combines both methods. Specifically, using LLaMA2-7B model as an example, we divide the pruning process into unstructure pruning and structure pruning. In the first stage, we use the SparseGPT method to prune weights. In the second stage, we compress the model obtained in the first stage by our proposed method. The experiment controlled the total pruning ratio at 37.5\%, with the proportion of structure and non-structure being adjusted. The results show that increasing SparseGPT’s pruning ratio while reducing our deep pruning ratio decreases perplexity (PPL) but reduces efficiency. Conversely, reducing PPL typically enhances efficiency.
> | Unstructured Ratio | Structured Ratio | PPL_WikiText2 |  PPL_PTB |  PPL_C4 | Latency(sec) | Throughout(tokens/s) |
> |:------------------:|:----------------:|:-------------:|:--------:|:-------:|:------------:|:--------------------:|
> |         0%         |       100%       |    38.1157    | 105.2407 | 28.7712 |    2.2141    |        57.8891       |
> |         28%        |        72%       |    24.7917    |  76.1376 | 20.0237 |    2.4163    |        52.9739       |
> |         50%        |        50%       |    18.2917    |  66.1394 | 16.1004 |    2.7269    |        46.9398       |
> |         72%        |        28%       |    14.4598    |  53.9301 | 12.9936 |     3.044    |        42.1448       |
> |        100%        |        0%        |    13.5921    |  50.4833 | 11.8936 |    3.3142    |        38.6315       |
>
> It is worth noting that during our experiments, we also discovered that, while ensuring approximate performance and efficiency requirements, a combined approach can achieve a higher compression rate compared to using a single pruning method. For instance, when we prune the model using 100\% structured pruning (without any unstructured pruning), the resulting model with approximately 5.5B parameters achieves perplexity (PPL) values of 18.8686, 61.8678, and 16.1392 on the WikiText2, PTB, and C4 datasets, respectively, with inference latency of 2.7554 seconds and throughput of 46.455 tokens/s. However, when we apply a combination of 26\% unstructured pruning and 74\% structured pruning, reducing the model to 4.6B parameters, we are able to maintain similar PPL values and inference efficiency.
> | params | Unstructured Ratio | Structured Ratio | PPL_WikiText2 | PPL_PTB |  PPL_C4 | Latency(sec) | Throughout(tokens/s) |
> |:------:|:------------------:|:----------------:|:-------------:|:-------:|:-------:|:------------:|:--------------------:|
> |  5.5B  |         0%         |       100%       |    18.8686    | 61.8678 | 16.1392 |    2.7554    |        46.455        |
> |  5.0B  |         14%        |        86%       |    18.3585    | 64.8451 | 15.6334 |    2.7678    |        46.249        |
> |  4.6B  |         26%        |        74%       |    18.2917    | 66.1394 | 16.1004 |    2.7269    |        46.9398       |
> |  4.1B  |         44%        |        56%       |    19.6860    | 63.9139 | 16.2838 |    2.7237    |        46.9953       |
>
> ---
> ## Question 4
> > Could the authors provide more interpretability on which types of layers (e.g., attention, feed-forward, early vs. late) are most frequently pruned, and relate this to observed task degradations?
>
> **Answer to Q4:** Thank you for your valuable suggestion. Our method, which views pruning from a game-theoretic perspective and uses Shapley values to measure layer importance, requires the players (i.e., the layers participating in the game) to have similar structural characteristics. For this reason, we have chosen Transformer layers as the smallest unit for pruning in our approach. We agree with the reviewer’s point about the importance of exploring different layer types, such as attention layers, feed-forward layers, and early vs. late layers, and how their pruning frequencies might impact task performance. This is an interesting direction, and we intend to investigate this in future work.

---

### Official Review · Reviewer_gSFA · 2025-10-31

**Soundness:** 3
**Presentation:** 3
**Contribution:** 3
**Rating:** 8
**Confidence:** 5

**Summary:**

This is a well-executed paper with a creative idea (cooperative game formulation + surrogate Shapley approximation) and extensive empirical validation. While more theoretical analysis of the surrogate approximation would strengthen the paper, the methodological novelty and practical effectiveness make it strong.

**Strengths:**

1. Viewing layer pruning as a cooperative game is original and well-motivated. It captures inter-layer dependencies often ignored in prior pruning methods based on static heuristics. The surrogate-assisted estimation is elegant and computationally practical, bridging theory and application.
2. Experiments are thorough, covering multiple models (transformer and non-transformer), datasets, and both generative and reasoning tasks, and generalization to quantization.
3. Consistently outperforms strong baselines (SliceGPT, SLEB, ShortGPT, Shortened-LLaMA) across tasks and pruning ratios. The improvements are meaningful, especially at high pruning levels.

**Weaknesses:**

1. While inspired by cooperative game theory, the connection remains mostly heuristic. The surrogate model approximates marginal contributions but lacks analysis of approximation error or variance bounds.
2. There is limited discussion of the surrogate’s accuracy or potential biases (e.g., overfitting to sampled masks). Reporting R² or correlation between predicted and true perplexities would strengthen the claim.
3. Although the method reduces evaluation costs compared to naive Shapley computation, 8k–80k mask evaluations and 200-epoch surrogate training are still substantial for large models. Quantitative runtime comparisons to baselines would help.

**Questions:**

1. What is the computational overhead (GPU hours) compared to simpler pruning baselines like ShortGPT or SliceGPT?
2. How sensitive is the method to the number of sampled masks or surrogate capacity?

---

> ### Author Response · Authors · 2025-11-23
> **Response to Reviewer gSFA**
>
> Thank you for the positive assessment and helpful feedback. We will address each of your comments in the following.
>
> ---
> ## Weakness 1
> > The surrogate model approximates marginal contributions but lacks analysis of approximation error or variance bounds.
>
> **Answer to W1:** Thank you for your valuable feedback. We evaluated the surrogate model's performance using the LLaMA-2-7B-hf model with the training mask ks=(30,27,24,21,18), corresponding to our predefined Hamming weight settings, and analyzed the R² metric. The results show that the surrogate model performs consistently across different random seeds and test sample sizes. For various mask configurations, we systematically varied the Hamming weight of the test mask. The results demonstrate that the surrogate model maintains high predictive accuracy when the test mask closely matches the training mask. However, when the test mask differs significantly from the training mask, the model’s performance declines, as expected. In our formal experiments, we will adjust the mask according to the desired pruning ratio, ensuring the test mask remains similar to the training mask. We have included the detailed experimental results in Appendix B.1 (Table 8) of the revised manuscript.
>
> |            Case            | Test Samples  | Seed  |         Test Mask          | R$^2$  |
> |:--------------------------|:-------------:|:-----:|:--------------------------:|:------:|
> | Same Seed, Same Mask  |      200      |   42  | ks = (30, 27, 24, 21, 18)  |  0.9360 |
> |                            |      500      |   42  | ks = (30, 27, 24, 21, 18)  | 0.9492 |
> |                            |      1000     |   42  | ks = (30, 27, 24, 21, 18)  | 0.9464 |
> | Different Seeds, Same Mask |      500      |  500  | ks = (30, 27, 24, 21, 18)  | 0.9359 |
> |                            |      500      |  1234 | ks = (30, 27, 24, 21, 18)  | 0.9402 |
> |                            |      500      | 99999 | ks = (30, 27, 24, 21, 18)  |  0.9400  |
> | Same Seed, Different Masks |      500      |   42  |  ks = (16, 27, 24, 21, 18) |  0.9230 |
> |                            |      500      |   42  |  ks = (16, 14, 24, 21, 18) | 0.8658 |
> |                            |      500      |   42  |  ks = (16, 14, 12, 21, 18) | 0.2466 |
>
> ---
> ## Weakness 2
> >  There is limited discussion of the surrogate’s accuracy or potential biases (e.g., overfitting to sampled masks). Reporting R² or correlation between predicted and true perplexities would strengthen the claim.
>
> **Answer to W2:** Thank you for your insightful feedback. As mentioned in Answer to W1, we have added the corresponding experimental results and updated the PDF. The experiments demonstrate the surrogate model's prediction accuracy and show that it performs well across different seed values, test sample sizes, and mask distributions.
>
> ---
> ## Weakness 3
> >  Although the method reduces evaluation costs compared to naive Shapley computation, 8k–80k mask evaluations and 200-epoch surrogate training are still substantial for large models. Quantitative runtime comparisons to baselines would help.
>
> **Answer to W3:** We have discussed the computational cost of our method in detail in Appendix G, with the results presented in Table 23. Specifically, for pruning the LLaMA-2-7B-hf model, the first step of generating 8000 masks for surrogate model training takes approximately 15 minutes on a 32GB V100 GPU. The second step, involving model training and using the surrogate model to simulate 80,000 masks for Shapley-based importance value computation, also takes about 15 minutes. For comparison, we tested SliceGPT and ShortGPT under the same experimental setup, where their pruning processes took 10 minutes and 2 minutes, respectively. Although our method is relatively more time-consuming, it is still within an acceptable range. Moreover, our method is theoretically well-founded and achieves better results.
>
> ---
> ## Question 1
> > What is the computational overhead (GPU hours) compared to simpler pruning baselines like ShortGPT or SliceGPT?
>
> **Answer to Q1:** Thank you for your insightful feedback. As mentioned in Answer to W3, our method takes approximately 30 minutes to prune the LLaMA-2-7B-hf model, while SliceGPT and ShortGPT complete the pruning in 10 minutes and 2 minutes, respectively, under the same experimental setup.
>
> ---
> ## Question 2
> > How sensitive is the method to the number of sampled masks or surrogate capacity?
>
> **Answer to Q2:** We have conducted various ablation experiments to evaluate the sensitivity of our method to factors such as the size and type of the Calibration Dataset, the Simulation Number for Layer Pruning, and the Hamming Weight Constraint for Mask Generation. The sensitivity to the number of sampled masks is discussed in Table 21, and further details on these experiments can be found in Appendix F. These results provide insights into how these parameters impact the model's performance and sensitivity.

---

> > ### Comment · Reviewer_gSFA · 2025-11-24
> >
> > The rebuttal has addressed my concerns, and i will maintain the score

---

> > > ### Author Response · Authors · 2025-11-26
> > >
> > > Dear Reviewer gSFA,
> > >
> > > We sincerely thank you for your detailed and insightful feedback, we are happy to see that our response address your concerns. Thanks again for your valuable comments to help us improve our paper.

---

### Author Response · Authors · 2025-11-23
**Update Manuscript**

We would like to thank all of the reviewers again for helping us improve the paper. We uploaded a revised version of our paper and marked the major modifications in blue for visibility. In short,

1. We add the MMLU benchmark in our evaluation to provide a broader view of the model’s capabilities.

2. We expand the discussion of relevant literature on research using game theory to provide a more comprehensive overview.

3. We revise the citation problem.

4. We add a discussion of Surrogate Model Performance in Appendix B.1.

5. We add an ablation study that further explores the impact of the Hamming weight constrain in Appendix F.3.

6. We add an experiment that integrates the unstructured pruning method and structured pruning method in Appendix H.

Thank you all again for your precious and insightful suggestions. Please let us know if you have additional questions or ideas for improvement.

Kind regards, Authors

---

### Author Response · Authors · 2025-12-03
**Summary of Rebuttal Updates and Reviewer Consensus**

Dear Program Chairs, Senior Area Chairs, Area Chairs, and Reviewers:

We sincerely thank you for your time and dedication throughout the review process. We greatly appreciate the constructive feedback from all reviewers, which has significantly helped us strengthen our manuscript. We understand that the changes in the AC assignment and review workflow have likely increased your burden. In this reply, we briefly summarize how our rebuttal addresses the major concerns raised and highlight the revisions made (which are also detailed in the General Reply).

---
**Reviewer gSFA gives a high score of 8** and mainly raised questions about technical details, including the computational cost of our method and its sensitivity to sampling masks. The reviewer also requested an analysis of the surrogate model’s accuracy. In response, we have expanded the manuscript with experimental analysis on the accuracy of the surrogate model, the sensitivity of the sampling mask, and the computational cost of our method. The reviewer expressed satisfaction with our response and kept the original score unchanged.

---
**Reviewer FgAt gives a score of 4**, with most of the comments requesting additional clarification on the method's effectiveness (such as accuracy measurement of the surrogate model, robustness of Hamming weight selection, and adequacy of hierarchical Monte Carlo mask sampling). In addition, the reviewer supplemented the relevant work and proposed modifications to the layout design. In our rebuttal, we provided supplementary experiments on surrogate model accuracy, expanded the ablation studies on Hamming weight selection and Monte Carlo sampling, and refined the discussion of related work. We further explored the integration scheme of our method with unstructured pruning methods through experiments. It is worth noting that the reviewer suggested to compare the fusion scheme with the SLEB method in the subsequent discussion. We have added this additional experiment, which further verifies the superiority of our method in the integration scheme. Based on this, we believe we’ve addressed most of the reviewer’s concerns.

---
**Reviewer CfBF assigns a score of 6**. The reviewer mainly suggested 1) increase the analysis of the surrogate model’s accuracy; 2) explain the selection of calibration dataset and the computational cost in our experiment; and 3) explore the fine-tuning after pruning. We have responded by providing additional experiments and explanations regarding the surrogate model’s accuracy, detailed our criteria for selecting the calibration dataset, and compared the computational costs of ShortGPT and SliceGPT. We also included fine-tuning experimental results in the appendix. We believe these revisions address all the reviewer’s concerns.

---
**Reviewer jT3z also assigns a score of 6**, focused on method design and suggested additional experiments with different evaluation criteria. In response, we further clarified our technical design, conducted specific comparative analysis with the literature proposed by the reviewers to highlight our contribution, and added additional experiments to directly address these issues.

---
We hope that the summary provided here will be helpful in supporting a fair and well-informed final decision. Thank you again for your time and consideration.

Best regards,

Authors of Paper 10392

---

### Meta-Review · Area_Chair_QsRJ · 2026-01-07

**Summary:**

The reviewers raised several concerns regarding the methodology and the theoretical backing of the proposed game-theoretic framework for layer pruning. Reviewer gSFA noted the originality of the approach but called for a more thorough theoretical analysis of the surrogate model's approximation capabilities. Reviewer FgAt highlighted the need for clearer validation of the surrogate model's predictive fidelity and the effectiveness of the adopted sampling strategy. Reviewer CfBF also questioned the stability of the Shapley estimates across different datasets and mentioned the limited diversity in the evaluation benchmarks.

**Reviewer Concerns:**

1. **Addressed Concerns**:
   - The authors expanded their experimental analysis to demonstrate the accuracy of the surrogate model, addressing the request for detailed evaluation metrics such as R^2 measures. This improvement was acknowledged by reviewer gSFA, who maintained a high score (8).
   - Additional experiments were provided in response to reviewer FgAt's request for analysis on surrogate model accuracy and sensitivity of sampling methods, leading to better clarity.

2. **Outstanding Concerns**:
   - Reviewer FgAt's comments regarding the need for rigorous quantitative evaluation of the surrogate’s prediction fidelity remain partially addressed.

   - The reviewers suggested further exploration of alternative pruning methods and comparisons with existing literature (e.g., SparseGPT, SLEB), indicating a need for enhanced validation and completeness in the experimental section.

**Reviewer Scores:**

- **Reviewer gSFA (8)**: Likely to maintain the score.

- **Reviewer FgAt (4)**: May have increased the score to 6 due to the expansion of the analytical content but still might have lingering concerns.

- **Reviewer CfBF (6)**: The score may have increased for addressing some but not all concerns regarding the robustness of the evaluation and assumptions made.

- **Reviewer jT3z (6)**: Likely to maintain or slightly increase the score based on the quality of new evidence provided.

---

### Decision · Program_Chairs · 2026-01-26

Accept (Poster)